# Predicting and elucidating the etiology of fatty liver disease: A machine learning modeling and validation study in the IMI DIRECT cohorts

**Naeimeh Atabaki-Pasdar**[1], **Mattias Ohlsson**[2,3], **Ana Viñuela**[4,5,6], **Francesca Frau**[7], **Hugo Pomares-Millan**[1], **Mark Haid**[8], **Angus G. Jones**[9], **E. Louise Thomas**[10], **Robert W. Koivula**[1,11], **Azra Kurbasic**[1], **Pascal M. Mutie**[1], **Hugo Fitipaldi**[1], **Juan Fernandez**[1], **Adem Y. Dawed**[12], **Giuseppe N. Giordano**[1], **Ian M. Forgie**[12], **Timothy J. McDonald**[9,13], **Femke Rutters**[14], **Henna Cederberg**[15], **Elizaveta Chabanova**[16], **Matilda Dale**[17], **Federico De Masi**[18], **Cecilia Engel Thomas**[17], **Kristine H. Allin**[19,20], **Tue H. Hansen**[19,21], **Alison Heggie**[22], **Mun-Gwan Hong**[17], **Petra J. M. Elders**[23], **Gwen Kennedy**[24], **Tarja Kokkola**[25], **Helle Krogh Pedersen**[19], **Anubha Mahajan**[26], **Donna McEvoy**[22], **Francois Pattou**[27], **Violeta Raverdy**[27], **Ragna S. Häussler**[17], **Sapna Sharma**[28,29], **Henrik S. Thomsen**[16], **Jagadish Vangipurapu**[25], **Henrik Vestergaard**[19,30], **Leen M. 't Hart**[14,31,32], **Jerzy Adamski**[8,33,34], **Petra B. Musholt**[35], **Soren Brage**[36], **Søren Brunak**[18,37], **Emmanouil Dermitzakis**[4,5,6], **Gary Frost**[38], **Torben Hansen**[19,39], **Markku Laakso**[25,40], **Oluf Pedersen**[19], **Martin Ridderstråle**[41], **Hartmut Ruetten**[7], **Andrew T. Hattersley**[9], **Mark Walker**[22], **Joline W. J. Beulens**[14,42], **Andrea Mari**[43], **Jochen M. Schwenk**[17], **Ramneek Gupta**[18], **Mark I. McCarthy**[11,26,44,45], **Ewan R. Pearson**[12], **Jimmy D. Bell**[10], **Imre Pavo**[46], **Paul W. Franks**[1,47] *

**Data Availability Statement:** Data cannot be shared publicly due to a need to maintain the confidentiality of patient data. Interested

1 Genetic and Molecular Epidemiology Unit, Department of Clinical Sciences, Lund University, Malmö, Sweden, 2 Computational Biology and Biological Physics Unit, Department of Astronomy and Theoretical Physics, Lund University, Lund, Sweden, 3 Center for Applied Intelligent Systems Research, Halmstad University, Halmstad, Sweden, 4 Department of Genetic Medicine and Development, University of Geneva Medical School, Geneva, Switzerland, 5 Institute for Genetics and Genomics in Geneva, University of Geneva Medical School, Geneva, Switzerland, 6 Swiss Institute of Bioinformatics, Geneva, Switzerland, 7 Sanofi-Aventis Deutschland, Frankfurt am Main, Germany, 8 Research Unit Molecular Endocrinology and Metabolism, Helmholtz Zentrum München, Neuherberg, Germany, 9 Institute of Biomedical and Clinical Science, College of Medicine and Health, University of Exeter, Exeter, United Kingdom, 10 Research Centre for Optimal Health, School of Life Sciences, University of Westminster, London, United Kingdom, 11 Oxford Centre for Diabetes, Endocrinology and Metabolism, Radcliffe Department of Medicine, University of Oxford, Oxford, United Kingdom, 12 Division of Population Health and Genomics, School of Medicine, University of Dundee, Ninewells Hospital, Dundee, United Kingdom, 13 Blood Sciences, Royal Devon and Exeter NHS Foundation Trust, Exeter, United Kingdom, 14 Department of Epidemiology and Biostatistics, Amsterdam Public Health Research Institute, Amsterdam UMC, Amsterdam, the Netherlands, 15 Department of Endocrinology, Abdominal Centre, Helsinki University Hospital, Helsinki, Finland, 16 Department of Diagnostic Radiology, Copenhagen University Hospital Herlev Gentofte, Herlev, Denmark, 17 Affinity Proteomics, Science for Life Laboratory, School of Engineering Sciences in Chemistry, Biotechnology and Health, KTH Royal Institute of Technology, Solna, Sweden, 18 Department of Health Technology, Technical University of Denmark, Kongens Lyngby, Denmark, 19 Novo Nordisk Foundation Center for Basic Metabolic Research, Faculty of Health and Medical Sciences, University of Copenhagen, Copenhagen, Denmark, 20 Center for Clinical Research and Prevention, Bispebjerg and Frederiksberg Hospital, Copenhagen, Denmark, 21 Department of Cardiology and Endocrinology, Slagelse Hospital, Slagelse, Denmark, 22 Institute of Cellular Medicine, Newcastle University, Newcastle upon Tyne, United Kingdom, 23 Department of General Practice, Amsterdam Public Health Research Institute, Amsterdam UMC, Amsterdam, the Netherlands, 24 Immunoassay Biomarker Core Laboratory, School of Medicine, University of Dundee, Ninewells Hospital, Dundee, United Kingdom, 25 Internal Medicine, Institute of Clinical Medicine, University of Eastern Finland, Kuopio, Finland, 26 Wellcome Centre for Human Genetics, University of Oxford, Oxford, United Kingdom, 27 University of Lille, Inserm, UMR 1190, Translational Research in Diabetes, Department of Endocrine Surgery, CHU Lille, Lille, France, 28 German Center for Diabetes Research, Neuherberg, Germany, 29 Unit of Molecular Epidemiology, Institute of Epidemiology II, Helmholtz Zentrum München, Neuherberg, Germany, 30 Steno Diabetes Center Copenhagen, Gentofte, Denmark, 31 Department of Cell and Chemical Biology, Leiden University Medical Center, Leiden, the Netherlands, 32 Molecular Epidemiology, Department of Biomedical Data Sciences, Leiden University Medical Center,

researchers may contact DIRECTdataaccess@dundee.ac.uk to request and obtain relevant data.

**Funding:** The work leading to this publication has received support from the Innovative Medicines Initiative Joint Undertaking under grant agreement n°115317 (DIRECT), resources of which are composed of financial contribution from the European Union's Seventh Framework Programme (FP7/2007-2013) and EFPIA companies' in kind contribution. NAP is supported in part by Henning och Johan Throne-Holsts Foundation, Hans Werthén Foundation, an IRC award from the Swedish Foundation for Strategic Research and a European Research Council award ERC-2015-CoG - 681742_NASCENT. HPM is supported by an IRC award from the Swedish Foundation for Strategic Research and a European Research Council award ERC-2015-CoG - 681742_NASCENT. AGJ is supported by an NIHR Clinician Scientist award (17/0005624). RK is funded by the Novo Nordisk Foundation (NNF18OC0031650) as part of a postdoctoral fellowship, an IRC award from the Swedish Foundation for Strategic Research and a European Research Council award ERC-2015-CoG - 681742_NASCENT. AK, PM, HF, JF and GNG are supported by an IRC award from the Swedish Foundation for Strategic Research and a European Research Council award ERC-2015-CoG - 681742_NASCENT. TJM is funded by an NIHR clinical senior lecturer fellowship. S.Bru acknowledges support from the Novo Nordisk Foundation (grants NNF17OC0027594 and NNF14CC0001). ATH is a Wellcome Trust Senior Investigator and is also supported by the NIHR Exeter Clinical Research Facility. JMS acknowledges support from Science for Life Laboratory (Plasma Profiling Facility), Knut and Alice Wallenberg Foundation (Human Protein Atlas) and Erling-Persson Foundation (KTH Centre for Precision Medicine). MIM is supported by the following grants; Wellcome (090532, 098381, 106130, 203141, 212259); NIH (U01-DK105535). PWF is supported by an IRC award from the Swedish Foundation for Strategic Research and a European Research Council award ERC-2015-CoG - 681742_NASCENT. The funders had no role in study design, data collection and analysis, decision to publish, or preparation of the manuscript.

**Competing interests:** I have read the journal's policy and the authors of this manuscript have the following competing interests: PWF is a consultant for Novo Nordisk, Lilly, and Zoe Global Ltd., and has received research grants from numerous diabetes drug companies. HR is an employee and shareholder of Sanofi. MIM: The views expressed

Leiden, the Netherlands, **33** Lehrstuhl für Experimentelle Genetik, Wissenschaftszentrum Weihenstephan für Ernährung, Landnutzung und Umwelt, Technische Universität München, Freising, Germany, **34** Department of Biochemistry, Yong Loo Lin School of Medicine, National University of Singapore, Singapore, **35** Diabetes Division, Research and Development, Sanofi, Frankfurt, Germany, **36** MRC Epidemiology Unit, University of Cambridge, Cambridge, United Kingdom, **37** Novo Nordisk Foundation Center for Protein Research, Faculty of Health and Medical Sciences, University of Copenhagen, Copenhagen, Denmark, **38** Section for Nutrition Research, Department of Metabolism, Digestion and Reproduction, Imperial College London, London, United Kingdom, **39** Faculty of Health Sciences, University of Southern Denmark, Odense, Denmark, **40** Department of Medicine, University of Eastern Finland and Kuopio University Hospital, Kuopio, Finland, **41** Clinical Pharmacology and Translational Medicine, Novo Nordisk, Søborg, Denmark, **42** Julius Center for Health Sciences and Primary Care, University Medical Center Utrecht, Utrecht, the Netherlands, **43** Institute of Neuroscience, National Research Council, Padua, Italy, **44** NIHR Oxford Biomedical Research Centre, Oxford University Hospitals NHS Foundation Trust, John Radcliffe Hospital, Oxford, United Kingdom, **45** OMNI Human Genetics, Genentech, South San Francisco, California, United States of America, **46** Eli Lilly Regional Operations, Vienna, Austria, **47** Department of Nutrition, Harvard School of Public Health, Boston, Massachusetts, United States of America

\* Paul.Franks@med.lu.se

# Abstract

## Background

Non-alcoholic fatty liver disease (NAFLD) is highly prevalent and causes serious health complications in individuals with and without type 2 diabetes (T2D). Early diagnosis of NAFLD is important, as this can help prevent irreversible damage to the liver and, ultimately, hepatocellular carcinomas. We sought to expand etiological understanding and develop a diagnostic tool for NAFLD using machine learning.

## Methods and findings

We utilized the baseline data from IMI DIRECT, a multicenter prospective cohort study of 3,029 European-ancestry adults recently diagnosed with T2D ($n$ = 795) or at high risk of developing the disease ($n$ = 2,234). Multi-omics (genetic, transcriptomic, proteomic, and metabolomic) and clinical (liver enzymes and other serological biomarkers, anthropometry, measures of beta-cell function, insulin sensitivity, and lifestyle) data comprised the key input variables. The models were trained on MRI-image-derived liver fat content (<5% or ≥5%) available for 1,514 participants. We applied LASSO (least absolute shrinkage and selection operator) to select features from the different layers of omics data and random forest analysis to develop the models. The prediction models included clinical and omics variables separately or in combination. A model including all omics and clinical variables yielded a cross-validated receiver operating characteristic area under the curve (ROCAUC) of 0.84 (95% CI 0.82, 0.86; $p < 0.001$), which compared with a ROCAUC of 0.82 (95% CI 0.81, 0.83; $p < 0.001$) for a model including 9 clinically accessible variables. The IMI DIRECT prediction models outperformed existing noninvasive NAFLD prediction tools. One limitation is that these analyses were performed in adults of European ancestry residing in northern Europe, and it is unknown how well these findings will translate to people of other ancestries and exposed to environmental risk factors that differ from those of the present cohort. Another key limitation of this study is that the prediction was done on a binary outcome of liver fat quantity (<5% or ≥5%) rather than a continuous one.

in this article are those of the author(s) and not necessarily those of the NHS, the NIHR, or the Department of Health. MIM has served on advisory panels for Pfizer, NovoNordisk and Zoe Global, has received honoraria from Merck, Pfizer, Novo Nordisk and Eli Lilly, and research funding from Abbvie, Astra Zeneca, Boehringer Ingelheim, Eli Lilly, Janssen, Merck, NovoNordisk, Pfizer, Roche, Sanofi Aventis, Servier, and Takeda. As of June 2019, MIM is an employee of Genentech, and a holder of Roche stock. AM is a consultant for Lilly and has received research grants from several diabetes drug companies.

**Abbreviations:** ALT, alanine transaminase; AST, aspartate transaminase; DBP, diastolic blood pressure; EFS, ensemble feature selection; FLI, fatty liver index; HBA1c, hemoglobin A1C; HSI, hepatic steatosis index; MMTT, mixed-meal tolerance test; MRI, magnetic resonance imaging; NAFLD, non-alcoholic fatty liver disease; NAFLD-LFS, non-alcoholic fatty liver disease liver fat score; NASH, non-alcoholic steatohepatitis; OGTT, oral glucose tolerance test; QC, quality control; ROCAUC, receiver operating characteristic area under the curve; SBP, systolic blood pressure; T2D, type 2 diabetes; TG, triglycerides.

## Conclusions

In this study, we developed several models with different combinations of clinical and omics data and identified biological features that appear to be associated with liver fat accumulation. In general, the clinical variables showed better prediction ability than the complex omics variables. However, the combination of omics and clinical variables yielded the highest accuracy. We have incorporated the developed clinical models into a web interface (see: https://www.predictliverfat.org/) and made it available to the community.

## Trial registration

ClinicalTrials.gov NCT03814915.

## Author summary

### Why was this study done?

- Globally, about 1 in 4 adults have non-alcoholic fatty liver disease (NAFLD), which adversely affects energy homeostasis (in particular blood glucose concentrations), blood detoxification, drug metabolism, and food digestion.

- Although numerous noninvasive tests to detect NAFLD exist, these typically include inaccurate blood-marker tests or expensive imaging methods.

- The purpose of this work was to develop accurate noninvasive methods to aid in the clinical prediction of NAFLD.

### What did the researchers do and find?

- The analyses applied machine learning methods to data from the deep-phenotyped IMI DIRECT cohorts ($n = 1,514$) to identify sets of highly informative variables for the prediction of NAFLD. The criterion measure was liver fat quantified from MRI.

- We developed a total of 18 prediction models that ranged from very inexpensive models of modest accuracy to more expensive biochemistry- and/or omics-based models with high accuracy.

- We found that models using measures commonly collected in either clinical settings or research studies proved adequate for the prediction of NAFLD.

- The addition of detailed omics data significantly improved the predictive utility of these models. We also found that of all omics markers, proteomic markers yielded the highest predictive accuracy when appropriately combined.

### What do these findings mean?

- We envisage that these new approaches to predicting fatty liver may be of clinical value when screening at-risk populations for NAFLD.

- The identification of specific molecular features that underlie the development of NAFLD provides novel insights into the disease's etiology, which may lead to the development of new treatments.

## Introduction

Non-alcoholic fatty liver disease (NAFLD) is characterized by the accumulation of fat in hepatocytes in the absence of excessive alcohol consumption. NAFLD is a spectrum of liver diseases, with its first stage, known as simple steatosis, defined as liver fat content ≥5% of total liver weight. Simple steatosis can progress to non-alcoholic steatohepatitis (NASH), fibrosis, cirrhosis, and eventually hepatocellular carcinoma. In NAFLD, triglycerides (TG) accumulate in hepatocytes, and liver insulin sensitivity is diminished, promoting hepatic gluconeogenesis, thereby raising the risk of type 2 diabetes (T2D) or exacerbating the disease pathology in those with diabetes [1–5]. Growing evidence also links an increased risk of cardiovascular events with NAFLD [6,7].

The prevalence of NAFLD is thought to be around 20%–40% in the general population in high-income countries, with numbers growing worldwide, imposing a substantial economic and public health burden [8–11]. However, the exact prevalence of NAFLD has not been clarified, in part because liver fat is difficult to accurately assess. Liver biopsy, magnetic resonance imaging (MRI), ultrasound, and liver enzyme tests are often used for NAFLD diagnosis, but the invasive nature of biopsies, the high cost of MRI scans, the non-quantitative nature and low sensitivity of conventional ultrasounds, and the low accuracy of liver enzyme tests are significant limitations [12–14]. To address this gap, several liver fat prediction indices have been developed, but none of these has sufficiently high predictive ability to be considered a gold standard [12].

The purpose of this study was to use machine learning to identify novel molecular features associated with NAFLD and combine these with conventional clinical variables to predict NAFLD. Our models include variables that are likely to be informative of disease etiology, some of which may be of use in clinical practice.

## Methods

### Participants (IMI DIRECT)

The primary data utilized in this study were generated within the IMI DIRECT consortium, which includes persons with diabetes ($n$ = 795) and without diabetes ($n$ = 2,234). All participants provided informed written consent, and the study protocol was approved by the regional research ethics committees for each clinical study center. Details of the study design and the core characteristics are provided elsewhere [15,16].

### Measures (IMI DIRECT)

A T2*-based multiecho technique was used to derive liver fat content from MRI [17,18], and the percentage values were categorized as fatty (≥5%) or non-fatty (<5%) to define the outcome variable. We elected not to attempt quantitative prediction of liver fat content, as this would require a much larger dataset to be adequately powered. A frequently-sampled 75-g oral glucose tolerance test (OGTT) or a frequently sampled mixed-meal tolerance test (MMTT) was performed, from which measures of glucose and insulin dynamics were calculated, as previously described [15,16]. Of 3,029 IMI DIRECT participants, 50% ($n$ = 1,514) had the liver fat

**Table 1. Characteristics of IMI DIRECT participants in the non-diabetes, diabetes, and combined cohorts separated for individuals with fatty liver versus non-fatty liver.**

| Characteristics | Non-diabetes cohort | | Diabetes cohort | | Combined cohort | |
|---|---|---|---|---|---|---|
| | Fatty liver | Non-fatty liver | Fatty liver | Non-fatty liver | Fatty liver | Non-fatty liver |
| N (percent) | 344 (34) | 667 (66) | 296 (59) | 207 (41) | 640 (42) | 874 (58) |
| Age (years) | 61 (56, 66) | 62 (56, 66) | 62 (55, 67) | 63 (58, 69) | 61 (56, 66) | 62 (56, 67) |
| Sex, n (percent female) | 62 (18) | 134 (20) | 130 (44) | 86 (42) | 192 (30) | 220 (25) |
| Weight (kg) | 90.75 (81.50, 100.25) | 81.40 (75.67, 89.60) | 92.85 (81.47, 103.75) | 80.80 (73.00, 93.55) | 91.20 (81.50, 102.00) | 81.40 (74.03, 90.17) |
| Waist circumference (cm) | 105 (98, 112) | 97 (91, 103) | 107 (97, 115) | 97 (90, 107) | 106 (98, 113) | 97 (91, 103) |
| BMI (kg/m$^2$) | 29.23 (26.91, 32.05) | 26.69 (24.75, 28.71) | 31.47 (28.37, 35.35) | 27.64 (25.53, 31.07) | 30.05 (27.53, 33.52) | 26.85 (24.91, 29.23) |
| SBP | 134.70 (125.30, 143.00) | 129.33 (120.00, 140.00) | 131 (122.00, 139.33) | 127.67 (117.67, 138.33) | 132.67 (124.00, 142.00) | 128.83 (119.33, 140.00) |
| DBP | 83.50 (79.33, 89.83) | 80.67 (75.67, 86.00) | 76.67 (72.00, 84.00) | 72.67 (67.17, 80.67) | 81.33 (5.33, 87.33) | 80.00 (73.33, 84.67) |
| HbA1c (mmol/mol) | 38 (36, 40) | 37 (35, 39) | 47 (44, 51) | 45 (42, 48) | 41 (37, 46) | 38 (36, 41) |
| Fasting glucose (mmol/l) | 5.90 (5.60, 6.30) | 5.70 (5.40, 6.00) | 7.20 (6.30, 7.90) | 6.70 (5.80, 7.60) | 6.30 (5.80, 7.20) | 5.80 (5.40, 6.30) |
| Fasting insulin (pmol/l) | 75.60 (54.30, 104.40) | 44.10 (27.75, 66.00) | 115.80 (75.80, 167.80) | 60.20 (40.85, 82.90) | 90.90 (61.20, 133.90) | 48.60 (30.00, 69.60) |
| 2-hour glucose (mmol/l) | 6.55 (5.37, 8.20) | 5.70 (4.70, 6.80) | 9.00 (6.90, 10.65) | 7.90 (6.20, 9.90) | 7.40 (5.90, 9.60) | 6.00 (4.90, 7.50) |
| 2-hour insulin (pmol/l) | 345.60 (198.40, 566.20) | 169.80 (100.20, 274.20) | 489.30 (297.40, 700.50) | 271.00 (166.40, 418.10) | 403.20 (236.60, 643.50) | 190.70 (110.80, 317.60) |
| Triglycerides (mmol/l) | 1.49 (1.13, 2.09) | 1.12 (0.86, 1.47) | 1.49 (1.01, 1.99) | 1.12 (0.86, 1.48) | 1.49 (1.08, 2.02) | 1.12 (0.86, 1.47) |
| ALT (units/l) | 21 (14, 29) | 15 (10, 20) | 25 (19, 33) | 20 (16, 24) | 23 (16, 32) | 16 (12, 22) |
| AST (units/l) | 29 (24, 37) | 25 (21, 30) | 24 (20, 30) | 22 (19, 27) | 26 (22, 33) | 24 (20, 29) |
| Alcohol intake, n for "never," "occasionally," "regularly" | 21, 68, 255 | 91, 133, 443 | 52, 81, 163 | 38, 45, 124 | 73, 149, 418 | 129, 178, 567 |
| Liver fat | 8.80 (6.60, 13.00) | 2.20 (1.50, 3.30) | 11.10 (7.30, 15.82) | 2.70 (1.95, 4.00) | 9.50 (6.80, 14.30) | 2.40 (1.60, 3.50) |

Values are median (interquartile range) unless otherwise specified.

ALT, alanine transaminase; AST, aspartate transaminase; BMI, body mass index; DBP, diastolic blood pressure; HbA1c, hemoglobin A1C; SBP, systolic blood pressure.

MRI data (503 with diabetes and 1,011 without diabetes). The distribution of the liver fat data among different centers and cohorts is shown in S1 and S2 Figs.

The list of the clinical input (predictor) variables (n = 58), including anthropometric measurements, plasma biomarkers, and lifestyle factors, are shown in S1 Table. These clinical variables were controlled for center effect by deriving residuals from a linear model including each clinical variable in each model; these residuals were then inverse normalized and used in subsequent analyses. Inverse normal transformation is a nonparametric method that replaces the data quantiles by quantiles from the standard normal distribution in order to reduce the impact of outliers and deviation from a normal distribution.

A detailed overview of participant characteristics for the key variables is shown in Table 1 for all IMI DIRECT participants with MRI data. There were no substantial differences in characteristics between these participants and those from IMI DIRECT who did not have MRI data (see S2 Table).

Genetic, transcriptomic, proteomic, and metabolomic datasets were used as input omics variables in the analyses. Buffy coat was separated from whole blood, and DNA was then extracted and genotyped using the Illumina HumanCore array (HCE24 v1.0); genotype imputation was performed using the Haplotype Reference Consortium (HRC) and 1000 Genomes (1KG) reference panels. Details of the quality control (QC) steps for the genetic data are

described elsewhere [16]. Transcriptomic data were generated using RNA sequencing from fasting whole blood. Only protein-coding genes were included in the analyses, as reads per kilobase of transcript per million mapped reads (RPKM). The targeted metabolomic data of fasting plasma samples were generated using the Biocrates AbsoluteIDQ p150 Kit. Additionally, untargeted LC/MS-based metabolomics was used to cover a broader spectrum of metabolites. A combination of technologies and quantitative panels of protein assays were used to generate "targeted" proteomic data. This included Olink proximity extension assays [19], sandwich immunoassay kits using Luminex technology (Merck Millipore and R&D Systems, Sweden), microfluidic ELISA assays (ProteinSimple, US [20]), protein analysis by Myriad RBM (Germany), and hsCRP analysis (MLM Medical Labs, Germany). In addition, protein data were generated by single-binder assays using highly multiplexed suspension bead arrays [21]. This approach (denoted "exploratory" proteomics) included a combination of antibodies targeting proteins selected by the consortium given published and unpublished evidence for association with glycemia-related traits. More information about data generation and QC of the transcriptomic, proteomic, and metabolomic data is provided in S1 Text. Technical covariates for transcriptomics include guanine-cytosine mean content, insert size, analysis lane and RNA integrity number, cell composition, date, and center. Technical covariates for proteomics were center, assay, plate number, and plate layout ($n = 4$), and for the targeted metabolites the technical covariates were center and plate. These technical covariates were used to correct the omics data, and the residuals were then extracted from these models and inverse normalized prior to further analyses.

## Feature selection (IMI DIRECT)

We developed a series of NAFLD prediction models composed of variables that are available within clinical settings, as well as those not currently available in most clinics (see S3 Table). We had 2 strategies for selecting the clinical variables. For models 1–3, we selected variables based on clinical accessibility and their established association with fatty liver from existing literature without applying statistical procedures for data reduction. For model 4, a pairwise Pearson correlation matrix was used for feature selection of the clinical variables by placing a pairwise correlation threshold of $r > 0.8$, and we then selected the variables we considered most accessible among those that were collinear. Feature selection was undertaken in the combined cohort (diabetes and non-diabetes) in order to maximize sample size and statistical power. Of 1,514 participants with liver fat data, 1,049 had all necessary clinical and multi-omics data for a complete case analysis. We used $k$-nearest neighbor [22] imputation with $k$ equal to 10 as a means to reduce the loss of sample size, but found that this did not materially improve predictive power in subsequent analyses, so we decided not to include these imputed data. An overview of the pairwise correlations among the clinical variables available in these 1,049 IMI DIRECT participants is presented in Fig 1.

The high-dimensionality nature of omics data also necessitated data reduction using the feature selection tool LASSO prior to building the model. LASSO is a regression analysis method that minimizes the sum of least squares in a linear regression model and shrinks selected beta coefficients ($\beta_j$) using penalties (Eq 1). Minimizing the value from Eq 1, LASSO excludes the least informative variables and selects those features of most importance for the outcome of interest ($y$) in a sample of $n$ cases, each of which consists of $m$ parameters. The penalty applied by $\lambda$ can be any value from 0 to positive infinity and is determined through a cross-validation step [26].

$$\sum_{i=1}^{n} (y_i - \hat{y}_i)^2 + \lambda \times \sum_{j=1}^{m} |\beta_j| \tag{1}$$

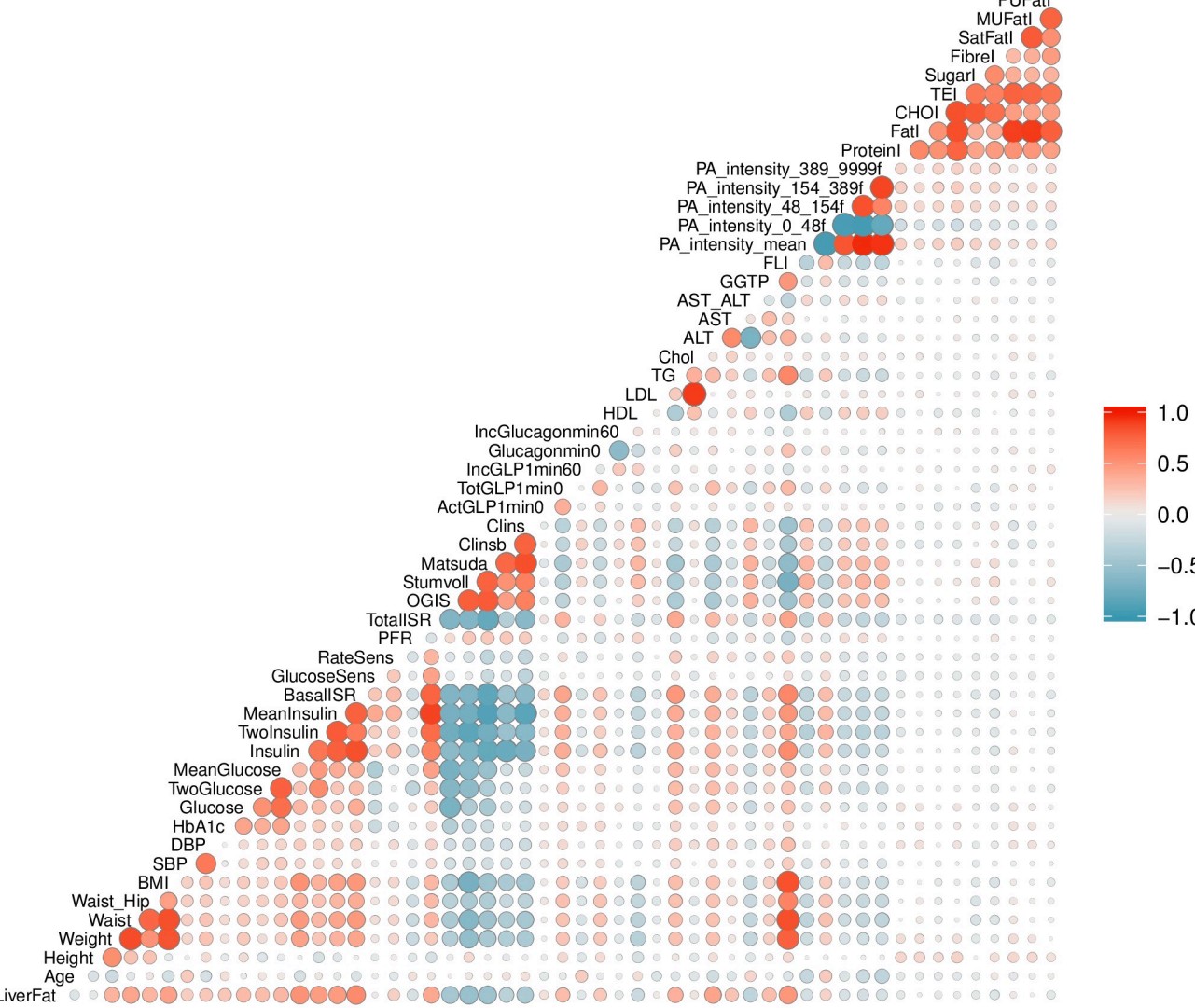

**Fig 1. Pearson pairwise correlation matrix of clinical variables (data are inverse normal transformed) in the cohort combining participants with and without diabetes in IMI DIRECT (*n* = 1,049).** The magnitude and direction of the correlation are reflected by the size (larger is stronger) and color (red is positive and blue is negative) of the circles, respectively. ActGLP1min0, concentration of fasting active GLP-1 in plasma; ALT, alanine transaminase; AST, aspartate transaminase; AST_ALT, AST to ALT ratio; BasalISR, insulin secretion at the beginning of the oral glucose tolerance test/mixed-meal tolerance test; BMI, body mass index; CHOI, total daily intake of dietary carbohydrates; Chol, total cholesterol; Clins, mean insulin clearance during the oral glucose tolerance test/mixed-meal tolerance test, calculated as (mean insulin secretion)/(mean insulin concentration); Clinsb, insulin clearance calculated from basal values as (insulin secretion)/(insulin concentration); DBP, mean diastolic blood pressure; FatI, total daily intake of dietary fats; FLI, fatty liver index; FibreI, total daily intake of dietary Association of Official Analytical Chemists (AOAC) fiber; GGTP, gamma-glutamyl transpeptidase; Glucagonmin0, fasting glucagon concentration; Glucose, fasting glucose from venous plasma samples; GlucoseSens, glucose sensitivity, slope of the dose–response relating insulin secretion to glucose concentration; HbA1c, hemoglobin A1C; HDL, fasting high-density lipoprotein cholesterol; IncGLP1min60, 1-hour GLP-1 increment; IncGlucagonmin60, 1-hour glucagon increment; Insulin, fasting insulin from venous plasma samples; LDL, fasting low-density lipoprotein cholesterol; Matsuda, insulin sensitivity index according to the method of Matsuda et al. [23]; MeanGlucose, mean glucose during the oral glucose tolerance test/mixed-meal tolerance test; MeanInsulin, mean insulin during the oral glucose tolerance test/mixed-meal tolerance test; MUFatI, daily intake of dietary monounsaturated fats; OGIS, oral glucose insulin sensitivity index according to the method of Mari et al. [24]; PA_intensity_0_48f, number of values in high-pass-filtered vector magnitude physical activity at ≥0 and ≤48; PA_intensity_154_389f, number of values in high-pass-filtered vector magnitude physical activity at ≥154 and ≤389; PA_intensity_389_9999f, number of values in high-pass-filtered vector magnitude physical activity at ≥389 and ≤9,999; PA_intensity_48_154f, number of values in high-pass-filtered vector magnitude physical activity at ≥48 and ≤154; PA_intensity_mean, mean high-pass-filtered vector magnitude physical activity intensity; PFR, potentiation factor ratio; ProteinI, total daily intake of dietary proteins; PUFatI, daily intake of dietary polyunsaturated fats; RateSens, rate sensitivity (parameter characterizing early insulin secretion); SatFatI, daily intake of dietary saturated fats; SBP, mean systolic blood pressure; Stumvoll, insulin sensitivity index according to the method of Stumvoll et al. [25]; SugarI, total daily intake of dietary; TEI, total daily energy intake based on validated multi-pass food habit questionnaire; TG, fasting triglycerides; TotalISR, integral of insulin secretion during the whole oral glucose tolerance test/mixed-meal tolerance test; TotGLP1min0, concentration of fasting total GLP-1 in plasma; TwoGlucose, 2-hour glucose after oral glucose tolerance test/mixed-meal tolerance test; TwoInsulin, 2-hour insulin; Waist_Hip, waist to hip ratio.

 

To minimize bias (for example by overfitting), we randomly divided the dataset and used 70% ($n = 735$) for feature selection and 30% ($n = 314$) for the model generation (see below). We selected these thresholds for partitioning the dataset in order to maximize the power to select the informative features. Stratified random sampling [27] based on the outcome variable was undertaken in order to preserve the distribution of the liver fat categories in the 2 feature selection and model generation sets. We selected LASSO, as a nonlinear data reduction tool might lead to overfitting owing to the high dimensionality of omics data. LASSO was conducted with package glmnet in R [28] with a 10-fold cross-validation step for defining the λ parameter that resulted in the minimum value for the mean square error of the regression model.

Feature selection using LASSO was undertaken in each omics dataset (genetic, transcriptomic, proteomic, and metabolomic) using 70% of the available data (models 5–18). For the genetic dataset, we first performed a genome-wide association study (GWAS) prior to LASSO in order to identify single nucleotide polymorphisms (SNPs) tentatively associated with liver fat accumulation ($p < 5 \times 10^{-6}$). LASSO was then applied to these index variants for feature selection in 70% of the study sample. The individual SNP association analysis was conducted with RVTESTS v2.0.2 [29], which applies a linear mixed model with an empirical kinship matrix to account for familial relatedness, cryptic relatedness, and population stratification. Only common variants with minor allele frequency (MAF) greater than 5% contributed to the kinship matrix. Liver fat data were log-transformed and then adjusted for age, age$^2$, sex, center, body mass index (BMI), and alcohol consumption. These values were then inverse normal transformed and used in the GWAS analyses. We limited our analysis to genetic MAF > 1% and imputation quality score > 0.3. S3 and S4 Figs show the resulting Manhattan plot, depicting each SNP's association with liver fat percentage and the quantile–quantile (QQ) plot of the GWAS results for liver fat. For the genetic data, 23 SNPs were selected out of the 108 SNPs with $p$-values $< 5 \times 10^{-6}$. For the transcriptomics, 93 genes were selected out of 16,209 protein-coding genes. In the exploratory and targeted proteomics, 22 out of 377 and 48 out of 483 proteins were selected, respectively. In the targeted and untargeted metabolomic data, 25 out of 116 and 39 out of 172 metabolites were selected by LASSO, respectively.

## Model training and evaluation

The remaining 30% of the data was used to develop the binary prediction models for fatty liver (yes/no) with selected features used as input variables. We utilized the random forest supervised machine learning method, which is an aggregation of decision trees built from bootstrapped datasets (a process called "bagging"). Typically, two-thirds of the data are retained in these bootstrapped datasets, and the remaining third is termed the out of bag (OOB) dataset, which is used to validate the performance of the model. To avoid overfitting and improve generalizability, 5-fold cross-validation was done for resampling the training samples and was repeated 5 times to create multiple versions of the folds. The number of trees was set to 1,000 to provide an accurate and stable prediction. Receiver operating characteristic (ROC) curves were used to evaluate model performance by measuring the area under the curve (AUC). A ROC curve uses a combination of sensitivity (true positive rate) and specificity (true negative rate) to assess prediction performance. In our analysis, the random forest model is used to derive probability estimates for the presence of fatty liver. In order to make a class prediction, it is necessary to impose a cutoff above which fatty liver is deemed probable and below which it is considered improbable. The choice of cutoff influences both sensitivity and specificity for a given prediction model. We considered the effect of different cutoffs on these performance measurements. Additionally, we calculated the F1 score, which is the harmonic mean of

 

precision (positive predictive value) and sensitivity, derived as follows:

$$\text{F1 score} = \frac{2 \times \text{sensitivity} \times \text{precision}}{(\text{sensitivity} + \text{precision})} \tag{2}$$

Balanced accuracy was also evaluated, which is the proportion of individuals correctly classified (true positives and true negatives) within each class individually. Measurements of sensitivity, specificity, F1 score, and balanced accuracy were computed and compared at different cutoffs for the diabetes, non-diabetes, and combined cohorts. The variable importance was also determined via a "permutation accuracy importance" measure using random forest analysis. In brief, for each tree, the prediction accuracy was calculated in the OOB test data. Each predictor variable was then permuted, and the accuracy was recalculated. The difference in the accuracies was averaged over all the trees and then normalized by the standard error. Thus, the measure for variable importance is the difference in prediction accuracy before and after the permutation for each variable [30]. In addition, we used the ensemble feature selection (EFS) method to determine the normalized importance value of all features [31]. With this approach, we do not rely on only random forest for the importance ranking, and we can build the cumulative importance values from different methods including Spearman's rank correlation test, Pearson's product moment correlation test, beta-values of logistic regression, the error-rate-based variable importance measure, and the Gini-index-based variable importance measure. Statistical analyses were undertaken using R software version 3.2.5 [32], and the random forest models were built using the caret package [33]. Fig 2 shows an overview of the different stages involved in the data processing and model training.

## Comparison with other fatty liver indices

Given the accessible data within the IMI DIRECT cohorts, several existing fatty liver indices could be calculated and compared with the IMI DIRECT prediction models. These included

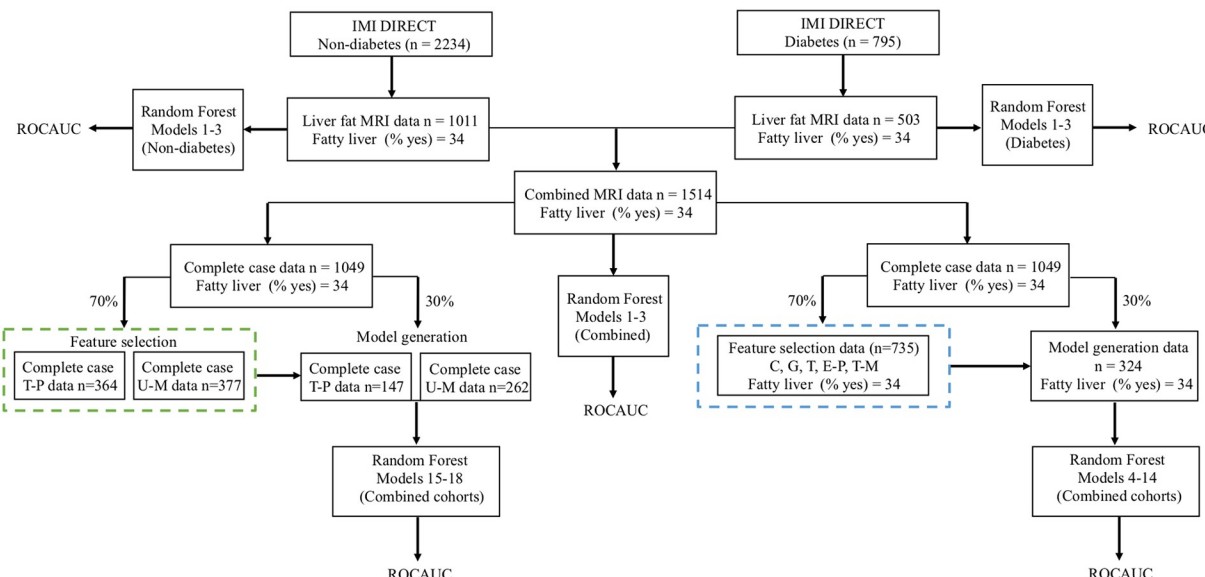

**Fig 2. Overview of the different stages involved in data processing and model training.** Data sources: clinical (C), genetic (G), transcriptomic (T), exploratory proteomic (E-P), targeted proteomic (T-P), targeted metabolomic (T-M), and untargeted metabolomic (U-M). The green and blue dashed boxes illustrate the feature selection step, the details of which can be found in S5 Fig. ROCAUC, receiver operating characteristic area under the curve.

the fatty liver index (FLI) [34], hepatic steatosis index (HSI) [35], and the NAFLD liver fat score (NAFLD-LFS) [36].

**FLI.** The FLI is commonly used to estimate the presence or absence of fatty liver (categorized into fatty [$\geq$60 FLI units] or non-fatty liver [<60 FLI units]) [34]. The FLI uses data on TG, waist circumference, BMI, and serum gamma-glutamyl transpeptidase (GGTP) and is calculated as follows:

$$\text{FLI} = \frac{e^{((0.953 \times \ln(TG)) + (0.139 \times BMI) + (0.718 \times \ln(GGTP)) + (0.053 \times Waist) - 15.745)}}{(1 + e^{((0.953 \times \ln TG)) + (0.139 \times BMI) + (0.718 \times \ln(GGTP)) + (0.053 \times Waist) - 15.745))}} \times 100 \qquad (3)$$

**NAFLD-FLS.** NAFLD-FLS was calculated using fasting serum (fs) insulin, aspartate transaminase (AST), alanine transaminase (ALT), T2D, and metabolic syndrome (MS) (defined according to the International Diabetes Federation [37]) to provide an estimate of liver fat content. A NAFLD-FLS value above −0.64 is considered to indicate the presence of NAFLD:

$$\text{NAFLD-LFS} = -2.89 + 1.18 \times MS \,(\text{yes } 1, \text{ no } 0) + 0.45 \times T2D \,(\text{yes } 2, \text{ no } 0)$$
$$+ 0.15 \times \text{fs Insulin} \qquad (4)$$

**HSI.** The HSI uses BMI, sex, T2D diagnosis (yes/no), and the ratio of ALT to AST and is calculated as follows:

$$\text{HSI} = 8 \times \frac{\text{ALT}}{\text{AST}} + \text{BMI}(+2 \text{ if T2D yes}, +2 \text{ if female}) \qquad (5)$$

HSI values above 36 are deemed to indicate the presence of NAFLD.

## External validation (UK Biobank cohort)

The UK Biobank cohort [38] was used to validate the clinical prediction models (models 1 and 2) derived using IMI DIRECT data (UK Biobank application ID: 18274). The same protocol and procedure have been used to quantify MRI-derived liver fat in IMI DIRECT and UK Biobank [18]. In addition, we validated the FLI and HSI using UK Biobank data. Field numbers for the UK Biobank variables used in the validation step can be found in the S4 Table. The data analysis procedures used for the UK Biobank validation analyses mirror those used in IMI DIRECT (as described above).

This study is reported as per the Strengthening the Reporting of Observational Studies in Epidemiology (STROBE) guideline (S1 STROBE Checklist) and the Transparent Reporting of a Multivariable Prediction Model for Individual Prognosis or Diagnosis (TRIPOD) guideline (S1 TRIPOD Checklist).

## Results

The following section describes fatty liver prediction models that are likely to suit different scenarios. We focus on a basic model (model 1), which includes variables that are widely available in both clinical and research settings. Models 2 and 3 focus on variables that could in principle be accessed within the clinical context, but that are not routinely available in the clinical setting at this time. Model 4 includes clinical variables, more detailed measures of glucose and insulin dynamics, and physical activity. Models 5 to 18 are more advanced models that include omics predictor variables alone or in combination with clinical predictor variables. See S3 Table for a full description of models.

## Clinical models 1–3

We developed models 1–3 for NAFLD prediction, graded by perceived data accessibility for clinicians. These models were developed on the full dataset without applying any statistical procedures for feature selection. Model 1 includes 6 non-serological input variables: waist circumference, BMI, systolic blood pressure (SBP), diastolic blood pressure (DBP), alcohol consumption, and diabetes status. Model 2 includes 8 input variables: waist circumference, BMI, TG, ALT, AST, fasting glucose (or hemoglobin A1C [HbA1c] if fasting glucose is not available), alcohol consumption, and diabetes status. Model 3 includes 9 variables: waist circumference, BMI, TG, ALT, AST, fasting glucose, fasting insulin, alcohol consumption, and diabetes status. Clinical models 1–3 along with the FLI, HSI, and NAFLD-LFS were applied to the non-diabetes and diabetes cohort datasets separately, as well as to the combined cohort dataset; the ROCAUC results are presented in Fig 3. Model 1 yielded a ROCAUC of 0.73 (95% CI 0.72, 0.75; $p < 0.001$) in the combined cohort. Adding serological variables to model 2 (with either fasting glucose or HbA1c) for the combined cohort yielded a ROCAUC of 0.79 (95% CI 0.78, 0.80; $p < 0.001$). Model 3 (fasting insulin added) yielded a ROCAUC of 0.82 (95% CI 0.81, 0.83; $p < 0.001$) in the combined cohort. The FLI, HSI, and NAFLD-LFS had ROCAUCs of 0.75 (95% CI 0.73, 0.78; $p < 0.001$), 0.75 (95% CI 0.72, 0.77; $p < 0.001$), and 0.79 (95% CI 0.76, 0.81; $p < 0.001$), respectively, in the combined cohort. The predictive performance of clinical models 1–3, FLI, HSI, and NAFLD-LFS in the non-diabetes and diabetes cohorts is presented in S5 Table.

## Performance metrics

We further investigated sensitivity, specificity, balanced accuracy, and F1 score (a score considering sensitivity and precision combined). These measurements were calculated for different cutoffs applied to the output of the random forest model (0.1, 0.2, 0.3, 0.4, 0.5, 0.6, 0.7, 0.8, and 0.9) using clinical models 1–3 in the diabetes, non-diabetes, and combined cohorts. The performance metrics for models 1 and 2 are presented in S6 and S7 Figs, and the metrics for model 3 are presented in Fig 4. We aimed to find the optimal cutoff for these models based on the cross-validated balanced accuracy. The highest balanced accuracy for models 1–3 in the non-diabetes, diabetes, and combined cohorts was observed at cutoffs of 0.4, 0.6, and 0.4, respectively (see Table 2).

Measurements of sensitivity, specificity, F1 score, and balanced accuracy were computed for the FLI, HSI, and NAFLD-LFS and compared with those of clinical models1–3. These measurements were computed at the optimal cutoff values for these indices: −0.640 for NAFLD-LFS, 60 for the FLI, and 36 for the HSI. A comprehensive overview of the prediction models' performance metrics for all of the fatty liver indices listed above is shown in Table 2.

## Validation in UK Biobank and IMI DIRECT

Liver fat data were available in 4,617 UK Biobank participants (1,011 with ≥5% liver fat and 3,606 with <5% liver fat). Of these individuals, 4,609 had all the required variables to replicate clinical model 1. To perform model 2, with either fasting glucose or HbA1c, 3,807 participants had data available for a complete case analysis. Given the limited availability of variables in the UK Biobank dataset, only models 1 and 2 of the NAFLD prediction models we developed could be externally validated. To facilitate this validation analysis, the random forest models developed in the IMI DIRECT cohorts were used to predict the liver fat category (participants with fatty liver versus non-fatty liver) for the UK Biobank participants. The performance of the FLI and HSI was also tested in the UK Biobank cohort. We validated both models 1 and 2 in the UK Biobank cohort with a similar ROCAUC as seen in the IMI DIRECT dataset. The

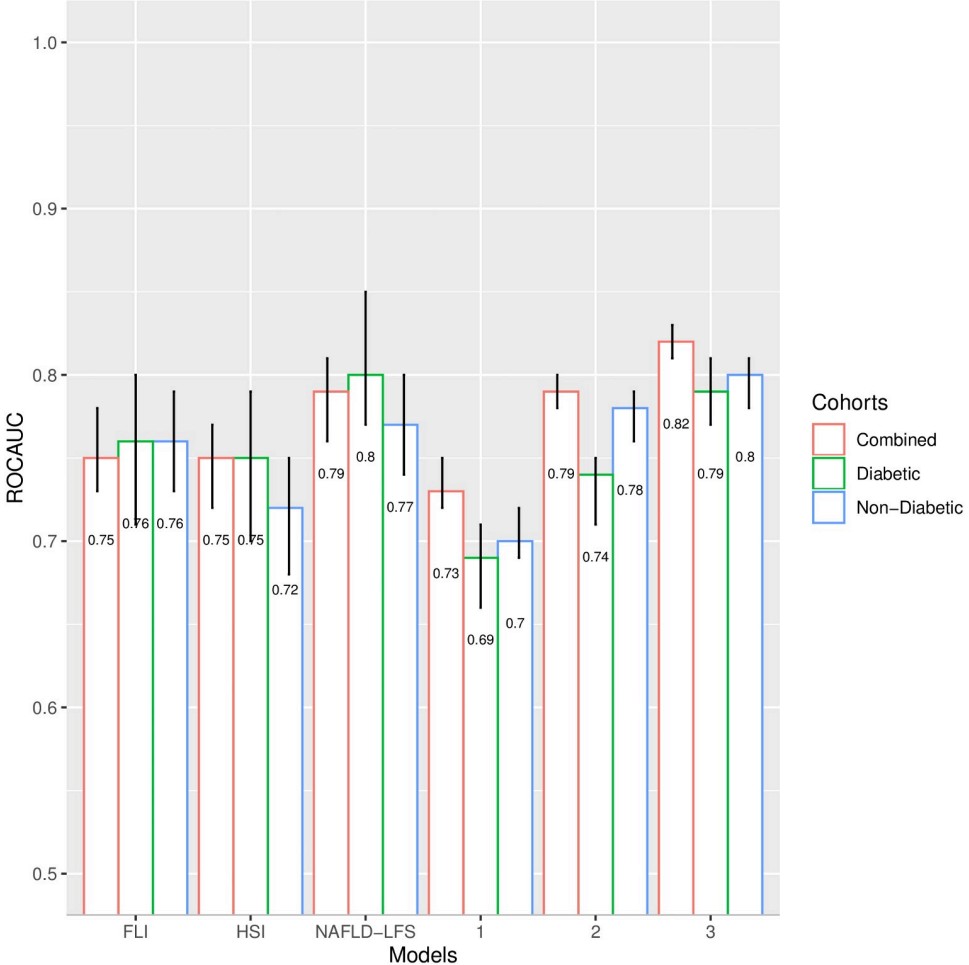

**Fig 3. Receiver operating characteristic area under the curve (ROCAUC) with 95% confidence interval (error bars) for clinical models 1–3, fatty liver index (FLI), hepatic steatosis index (HSI), and non-alcoholic fatty liver disease liver fat score (NAFLD-LFS) in the IMI DIRECT cohorts.** Model 1 includes 6 non-serological input variables: waist circumference, body mass index(BMI), mean systolic blood pressure, mean diastolic blood pressure, alcohol consumption, and diabetes status. Model 2 includes 8 input variables: waist circumference, BMI, fasting triglycerides (TG), alanine transaminase (ALT), aspartate transaminase (AST), fasting glucose (or hemoglobin A1C if fasting glucose is not available), alcohol consumption, and diabetes status. Model 3 includes 9 variables: waist circumference, BMI, TG, ALT, AST, fasting glucose, fasting insulin, alcohol consumption, and diabetes status. The FLI uses TG, waist circumference, BMI, and gamma-glutamyl transpeptidase. NAFLD-FLS was calculated using fasting insulin, AST, ALT, type 2 diabetes (T2D), and metabolic syndrome defined according to the International Diabetes Federation. The HSI uses BMI, sex, T2D diagnosis (yes/no), and the ratio of ALT to AST.

ROCAUCs were 0.71 (95% CI 0.69, 0.73; $p < 0.001$), 0.79 (95% CI 0.77, 0.80; $p < 0.001$), and 0.78 (95% CI 0.76, 0.79; $p < 0.001$) for model 1, model 2 with fasting glucose, and model 2 with HbA1c, respectively. The FLI had a ROCAUC of 0.78 (95% CI 0.76, 0.80; $p < 0.001$), which is similar to the ROCAUC of model 2. The HSI yielded a ROCAUC of 0.76 (95% CI 0.75, 0.78; $p < 0.001$).

Measurements of sensitivity, specificity, F1 score, and balanced accuracy were also computed at the optimal cutoff values for these models: 0.4 for clinical models 1 and 2, 60 for the FLI, and 36 for the HSI (see Table 2).

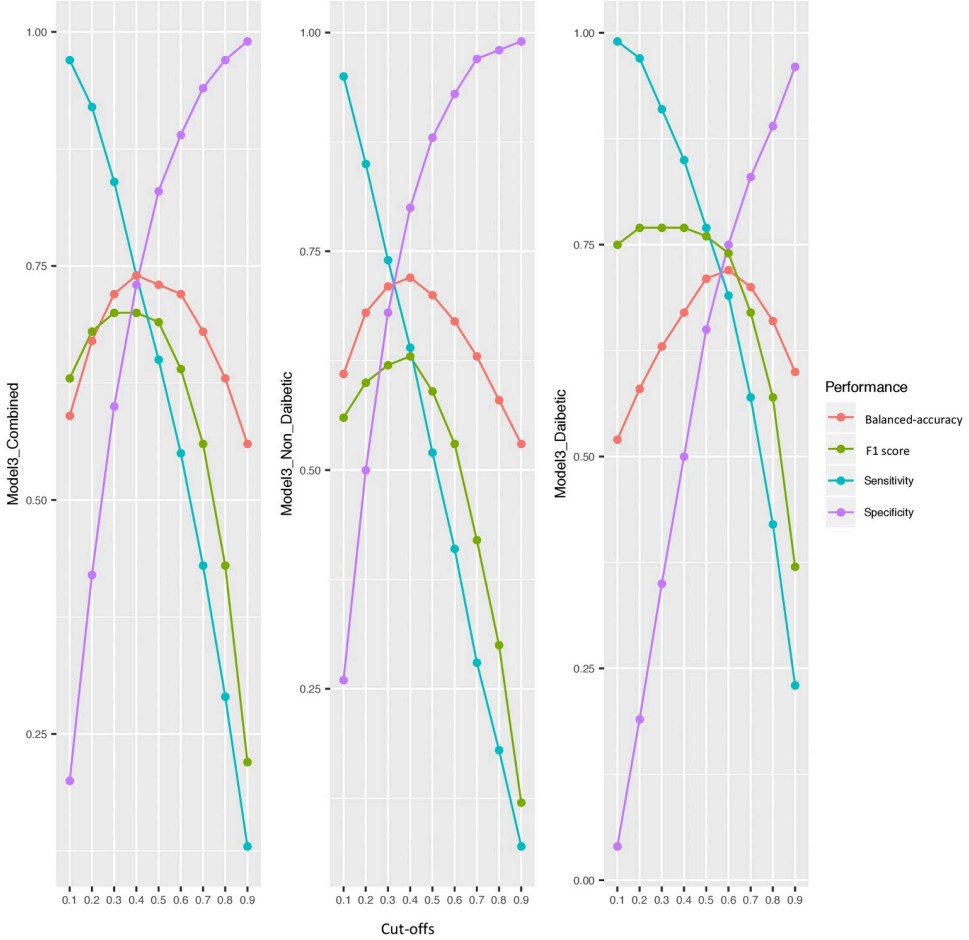

**Fig 4. Measurements of sensitivity, specificity, F1 (a score considering sensitivity and precision combined), and balanced accuracy at different cutoffs for model 3 in the diabetes, non-diabetes, and combined cohorts of IMI-DIRECT.** The measurements are calculated by defining the predicted probabilities of fatty liver equal to or above these cutoffs as fatty liver, and below as non-fatty liver. Model 3 includes 9 variables: waist circumference, body mass index, fasting triglycerides, alanine transaminase, aspartate transaminase, fasting glucose, fasting insulin, alcohol consumption, and diabetes status.

## Clinical model 4 and omics models 5–14

More advanced models using omics data were also developed. These models were generated using the omics features selected by LASSO in the combined cohort. The models include only omics or include omics plus 22 clinical variables as the input variables. Twenty-one of these clinical variables were selected based on the pairwise Pearson correlation matrix: BMI, waist circumference, SBP, DBP, alcohol consumption, ALT, AST, GGTP, HDL, TG, fasting glucose, 2-hour glucose, HbA1c, fasting insulin, 2-hour insulin, insulin secretion at the beginning of the carbohydrate challenge test (OGTT or MMTT), 2-hour oral glucose insulin sensitivity index (OGIS), mean insulin clearance during the OGTT/MTT, fasting glucagon concentration, fasting plasma total GLP-1 concentration, and mean physical activity intensity. Diabetes status (non-diabetes/diabetes) was also included as a clinical predictor in the models, given that analyses were undertaken in the combined diabetes and non-diabetes cohort. The ROCAUCs for models 4–14 are shown in Fig 5. The clinical model with the 22 selected clinical

**Table 2. An overview of the prediction models' performance metrics for clinical models 1–3, fatty liver index (FLI), hepatic steatosis index (HIS), and non-alcoholic fatty liver disease liver fat score (NAFLD-LFS) in the IMI DIRECT and UK Biobank datasets.**

| Cohort and model | Cutoff | Sensitivity | Specificity | F1 score | Balanced accuracy |
|---|---|---|---|---|---|
| **Non-diabetes (IMI DIRECT)** | | | | | |
| Model 1 | 0.4 | 0.51 | 0.75 | 0.51 | 0.63 |
| Model 2 | 0.4 | 0.60 | 0.79 | 0.59 | 0.69 |
| Model 3 | 0.4 | 0.64 | 0.80 | 0.63 | 0.72 |
| FLI | 60 | 0.89 | 0.41 | 0.58 | 0.65 |
| HSI | 36 | 0.62 | 0.68 | 0.55 | 0.65 |
| NAFLD-LFS | −0.64 | 1 | 0.04 | 0.51 | 0.52 |
| **Diabetes (IMI DIRECT)** | | | | | |
| Model 1 | 0.6 | 0.63 | 0.64 | 0.67 | 0.64 |
| Model 2 | 0.6 | 0.65 | 0.68 | 0.69 | 0.67 |
| Model 3 | 0.6 | 0.69 | 0.75 | 0.74 | 0.72 |
| FLI | 60 | 0.77 | 0.54 | 0.73 | 0.66 |
| HSI | 36 | 0.83 | 0.48 | 0.75 | 0.65 |
| NAFLD-LFS | −0.64 | 1 | 0.01 | 0.73 | 0.50 |
| **Combined (IMI DIRECT)** | | | | | |
| Model 1 | 0.4 | 0.67 | 0.65 | 0.62 | 0.66 |
| Model 2 | 0.4 | 0.72 | 0.69 | 0.67 | 0.71 |
| Model 3 | 0.4 | 0.74 | 0.73 | 0.70 | 0.74 |
| FLI | 60 | 0.84 | 0.44 | 0.64 | 0.64 |
| HSI | 36 | 0.71 | 0.63 | 0.64 | 0.67 |
| NAFLD-LFS | −0.64 | 1 | 0 | 0.58 | 0.50 |
| **UK Biobank** | | | | | |
| Model 1 | 0.4 | 0.49 | 0.78 | 0.43 | 0.63 |
| Model 2 | 0.4 | 0.67 | 0.74 | 0.52 | 0.71 |
| FLI | 60 | 0.62 | 0.76 | 0.50 | 0.69 |
| HSI | 36 | 0.66 | 0.72 | 0.50 | 0.69 |

Model 1 includes 6 non-serological input variables: waist circumference, body mass index (BMI), mean systolic blood pressure, mean diastolic blood pressure, alcohol consumption, and diabetes status. Model 2 includes 8 input variables: waist circumference, BMI, fasting triglycerides (TG), alanine transaminase (ALT), aspartate transaminase (AST), fasting glucose (or hemoglobin A1C if fasting glucose is not available), alcohol consumption, and diabetes status. Model 3 includes 9 variables: waist circumference, BMI, TG, ALT, AST, fasting glucose, fasting insulin, alcohol consumption, and diabetes status. The FLI uses TG, waist circumference, BMI, and gamma-glutamyl transpeptidase. NAFLD-FLS was calculated using fasting insulin, AST, ALT, type 2 diabetes (T2D), and metabolic syndrome defined according to the International Diabetes Federation. The HSI uses BMI, sex, T2D diagnosis (yes/no), and the ratio of ALT to AST.

variables (model 4) yielded a ROCAUC of 0.79 (95% CI 0.76, 0.81; $p < 0.001$). Omics models with only the genetic (model 5), transcriptomic (model 7), proteomic (model 9), and targeted metabolomic (model 11) data as input variables resulted in ROCAUCs of 0.67 (95% CI 0.65, 0.70; $p < 0.001$), 0.72 (95% CI 0.69, 0.74; $p < 0.001$), 0.74 (95% CI 0.71, 0.76; $p < 0.001$), and 0.70 (95% CI 0.67, 0.72; $p < 0.001$), respectively. Including all the omics variables in one model (model 13) resulted in a ROCAUC of 0.82 (95% CI 0.80, 0.84; $p < 0.001$). Adding the clinical variables to each omics model improved the prediction ability; models with the clinical variables plus genetic (model 6), transcriptomic (model 8), exploratory proteomic (model 10), and targeted metabolomic (model 12) data resulted in ROCAUCs of 0.82 (95% CI 0.80, 0.84; $p < 0.001$), 0.81 (95% CI 0.79, 0.83; $p < 0.001$), 0.80 (95% CI 0.78, 0.83; $p < 0.001$), and 0.80 (95% CI 0.77, 0.82; $p < 0.001$), respectively. The highest performance was observed for model 14 (ROCAUC of 0.84; 95% CI 0.82, 0.86; $p < 0.001$). The variable importance for model 14 from

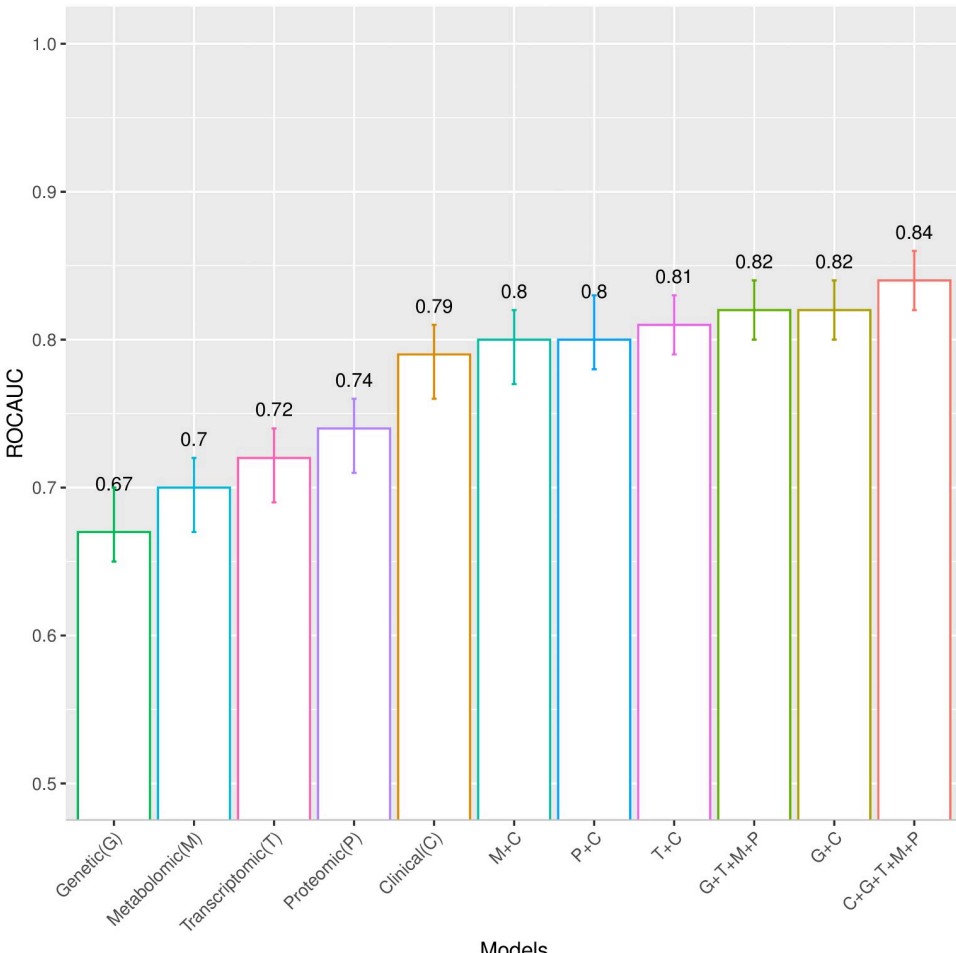

**Fig 5. Receiver operating characteristic area under the curve (ROCAUC) with 95% confidence interval for the clinical model and the omics separately or in combination with the clinical model in the IMI DIRECT combined cohort.** Clinical (C), model 4, with the 22 selected clinical variables. Genetic (G), model 5, with 23 SNPs. C+G, model 6, with clinical plus genetic variables. Transcriptomic (T), model 7, with 93 protein-coding genes. T+C, model 8, with transcriptomic plus clinical variables. Proteomic (P), model 9, with 22 proteins from exploratory proteomics. P+C, model 10, with proteomic plus clinical variables. Metabolomic (M), model 11, with 25 metabolites from targeted metabolomics. M+C, model 12, with metabolomic plus clinical variables. G+T+M+P, model 13, with all omics together. C+G+T+M+P, model 14, with all the omics combined with the clinical model.

the permutation accuracy importance measure, presented in Fig 6, shows that measures of insulin secretion rank amongst those having the highest variable importance of all input variables. Moreover, the importance list derived from EFS, shown in S20 Fig, is highly consistent with that derived from the random forest analysis. Rankings for the individual clinical and omics variables from the permutation accuracy importance measure and EFS are presented in S8–S19 Figs. The minor inconsistencies in results from the 2 approaches are likely to reflect the ability of the random forest analysis to detect variables that interact with others, which the linear methods are not designed to detect.

## Additional proteomic and metabolomic analyses (models 15–18)

Data from targeted proteomic and untargeted metabolomic data were further utilized to develop the omics models separately or in combination with the clinical data. However,

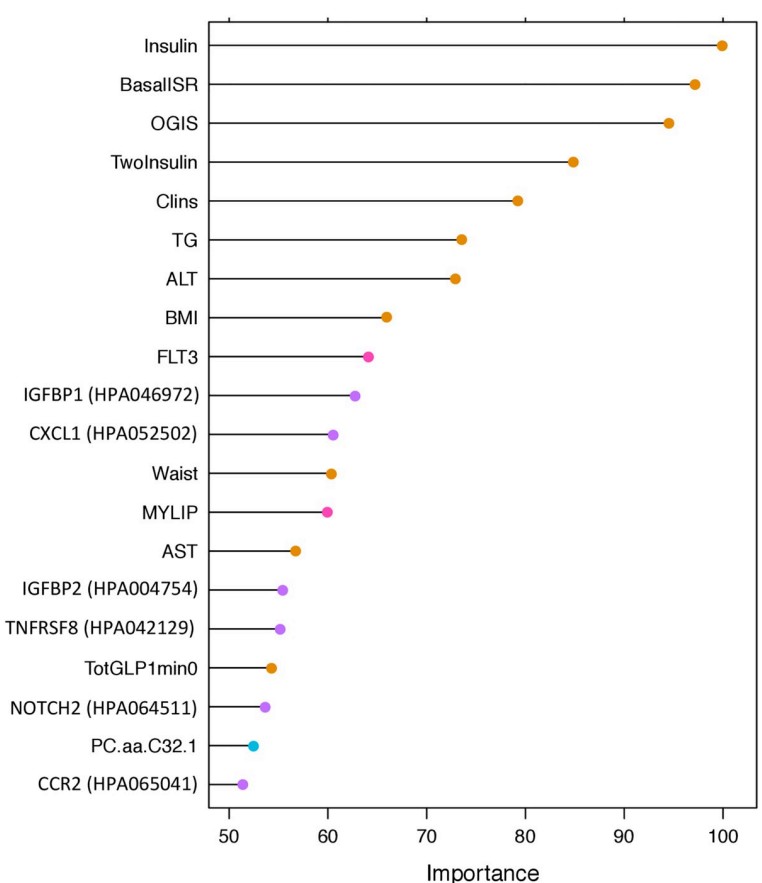

**Fig 6. Variable importance for the advanced model 14 with 185 omics and clinical input variables (clinical = 22, genetic = 23, transcriptomic = 93, exploratory proteomic = 22, and targeted metabolomic = 25).** The *y*-axis shows the top 20 predictors in the model. The *x*-axis shows the variable importance calculated, via a permutation accuracy importance measure using random forest analysis, as the difference in prediction accuracy before and after the permutation for each variable scaled by the standard error. ALT, alanine transaminase; AST, aspartate transaminase; BasalISR, insulin secretion at the beginning of the oral glucose tolerance test/mixed-meal tolerance test; BMI, body mass index; Clins, mean insulin clearance during the oral glucose tolerance test/mixed-meal tolerance test calculated as (mean insulin secretion)/(mean insulin concentration); FLT3, fetal liver tyrosine kinase-3; Insulin, fasting insulin from venous plasma samples; MYLIP, myosin regulatory light chain interacting protein; OGIS, oral glucose insulin sensitivity index according to the method of Mari et al. [24]; TG, fasting triglycerides; TotGLP1min0, concentration of fasting total GLP-1 in plasma; TwoInsulin, 2-hour insulin after oral glucose tolerance test/mixed meal tolerance test.

as some participants lacked these omics data, their models were developed using a smaller data subset and were, hence, not included in the advanced (model 14) analyses. The complete case analysis was primarily defined on the availability of the 22 selected clinical variables (*n* = 1,049). Within this complete case set, 511 had a complete set of untargeted metabolomic data, and 686 had a complete set of targeted proteomic data. The models with targeted proteomic data only and with proteomic and clinical variables combined resulted in ROCAUCs of 0.81 (95% CI 0.78, 0.84; $p < 0.001$) and 0.84 (95% CI 0.81, 0.87; $p < 0.001$), respectively. The untargeted metabolomic model alone had a ROCAUC of 0.66 (95% CI 0.63, 0.69; $p < 0.001$), which increased to 0.78 (95% CI 0.75, 0.80; $p < 0.001$) when the 22 clinical variables were added.

## Discussion

Using data from the IMI DIRECT consortium, we developed 18 diagnostic models for early-stage NAFLD. These models were developed to reflect different scenarios within which they might be used: These included both clinical and research settings, with the more complex (and less accessible) models having the greatest predictive ability. The models were successfully validated in the UK Biobank where data permitted such analysis (clinical models 1 and 2). Overall, the basic clinical variables proved to be stronger predictors of fatty liver than more complex omics data, although adding omics data yielded the most powerful model, with very good cross-validated predictive ability (ROCAUC = 0.84).

NAFLD is etiologically complex, rendering its prevention and treatment difficult, and diagnosis can require invasive and/or relatively expensive procedures. Thus, noninvasive and cost-effective prediction models with good sensitivity and specificity are much needed. This is especially important because if NAFLD is detected early, treatment through lifestyle interventions can be highly effective [39]. However, simple steatosis is usually asymptomatic, and many patients only come to the attention of hepatologists when serious complications arise [40].

To date, several prediction models have been developed to facilitate the diagnosis of steatosis (thoroughly reviewed elsewhere [13]). The FLI is one of the most well-established and commonly used fatty liver indices, initially developed using ultrasound-derived hepatic steatosis data [34]. The FLI yielded similar predictive performance in the diabetes and non-diabetes cohorts of IMI DIRECT (both ROCAUCs approximately 0.75).

Though commonly used for liver fat prediction, the FLI has a similar discriminative ability as waist circumference alone [41]. Better discrimination can be obtained by incorporating additional serological and hemostatic measures, which is the case with NAFLD-LFS [14], the SteatoTest [42], and the HSI [35], for example. Notwithstanding the added complexity and cost of these scores, the FLI, HSI, and NAFLD-LFS yielded similar predictive ability in a series of liver-biopsy-diagnosed NAFLD cases ($n$ = 324) [36].

Omics technologies have been used in a small number of studies to identify molecular biomarkers of NAFLD [43–45]. These include tests utilizing genetic data such as FibroGENE for staging liver fibrosis [46], and tests using metabolomic data derived from liver tissue to differentiate simple hepatitis from NASH [47], as well as a multi-component NAFLD classifier using genomic, proteomic, and phenomic data [45]. Machine learning models based on lipidomic, glycomic, and free fatty acid data were also developed for the diagnosis of NASH and liver fibrosis [48,49]. In a recent retrospective case series of patients with obesity, EFS was applied for feature selection, using a set of sociodemographic and serum variables to predict the presence or absence of NASH [50].

Using data from IMI DIRECT, we explored the predictive ability of genetic, transcriptomic, proteomic, and metabolomic data from blood in the diagnosis of NAFLD. The top 20 features of each omics model are presented in S9–S14 Figs. The details of the LASSO selected features are summarized in S7 Table. Reassuringly, several of the features that ranked highest have been previously described for their association with liver fat content or closely related traits; these include *PNPLA3* gene variants [44,51], fetal liver tyrosine kinase-3 (FLT3) transcripts [52], IGFBP1 [53–55] and lipoprotein lipase (Lpl) [56] proteins, and the metabolite glutamate [57]. In the analysis of the targeted metabolites, phosphatidylcholines (including PC.aa.C32, PC.aa.C38, PC.aa.C40, and PC.aa.C42), glycerophospholipids, and valine were amongst the highest-ranked metabolites that are known for their correlation with NAFLD and metabolic disorders [58,59]. For exploratory proteomics, the most important variables were proteins secreted into the blood, expressed by the liver as well as those leaking from the blood cells [60]. The prediction model that only included targeted proteomic data (model 15) performed well

(ROCAUC = 0.81), rendering it an interesting candidate biomarker for future clinical tests. Among the top 20 most important proteins were many secreted into the blood or leaked by the liver, as well as the pancreas, fat, or muscle tissue [61].

Our intention by including all features in the same model (model 14) was to maximize predictive power by leveraging interactions between features. Moreover, we explored the value of boosting ensemble algorithms for each data source. The purpose of this was to enhance predictions. We trained a stochastic gradient boosting algorithm for each data source separately and then applied a weighted averaging on the probabilities of observations. The optimal weighting was observed at 0.5 for the clinical data and 0.125 for each omics data layer (i.e., genetic, transcriptomic, exploratory proteomic, and targeted metabolomic). The ensemble prediction model of omics and clinical datasets resulted in a ROCAUC of 0.83 (95% CI 0.78, 0.87; $p <$ 0.001), which is not materially different from the ROCAUC derived for the advanced model 14 (described in the Results), which includes all the omics and clinical features in a single model (ROCAUC of 0.84; 95% CI 0.82, 0.86; $p < 0.001$). The models developed here may be useful for screening for NAFLD, and this should be evaluated in future clinical studies.

In order to stratify people into groups of those unlikely and likely to have NAFLD, the latter of whom might subsequently undergo more invasive and/or costly clinical assessments, it would be important for the prediction model to have high sensitivity. However, the predictive utility of a given model can be further improved by selecting model cutoffs that optimize sensitivity or specificity, as the 2 metrics rarely perform optimally at the same cutoff. This issue was apparent for models 1–3 in the current analyses, where we selected cutoffs that maximized balanced accuracy (considering both sensitivity and specificity); these features are especially important in screening algorithms, where the cost of false negatives can be high. Models 1–3 resulted in higher sensitivity in the diabetes cohort than the non-diabetes cohort, whereas the specificity was higher in the non-diabetes and combined cohorts than in the diabetes cohort.

The linear LASSO method was used to minimize overfitting that can occur with high-dimensionality data, while random forest analysis was used to identify nonlinear associations where data structure allowed. We also considered several other machine learning approaches including generalized linear model, stochastic gradient boosting, support vector machines, and $k$-nearest neighbor, and the random forest analysis yielded similar or better results compared with any of these other approaches (see S6 Table).

A limitation of the analytical approach used here is that the methods required a complete case analysis, which diminishes sample size considerably; although imputing missing data here helped preserve sample size, it did not improve the prediction ability of the models, and we hence elected to use the complete case analysis. Heavy alcohol consumption is a key determinant of fatty liver, but is unlikely to be a major etiological factor in IMI DIRECT owing to the demographics of this cohort. Nevertheless, a further limitation of this analysis is that alcohol intake was self-reported and may lack validity. To address this limitation, we removed all self-reported heavy alcohol consumers from the UK Biobank cohort and undertook sensitivity analyses, but this did not materially affect the results.

Here we considered lifestyle variables, but not medications. The use of medicines affecting liver fat is likely to be less in the non-diabetes than in the diabetes cohort, yet the models fit better in the latter, suggesting that glucose-lowering medication use in the IMI DIRECT cohorts did not have a major detrimental impact on prediction model performance.

A further consideration for future work is the impact lifestyle and medications are likely to have on the prediction of NAFLD. Furthermore, this study was undertaken in people of European ancestry, and the extent to which the results will generalize to other ethnic groups is unknown. Moreover, the prediction is for a binary liver fat outcome ($<$5% or $\geq$5%), and

neither fully quantifies liver fat volume nor elucidates the degree of liver damage (cirrhosis). These key limitations of the current work will be the focus of future research.

Our finding that a model focused on proteomic data yielded high predictive utility may warrant further investigation. Our analysis also suggests that insulin sensitivity and beta-cell dysfunction may be involved in liver fat accumulation, which are at present not considered as features of conventional NAFLD risk models.

In summary, we have developed prediction models for NAFLD that may have utility for clinical diagnosis and research investigations alike. A web interface for the diagnosis of NAFLD was developed using the findings described above (https://www.predictliverfat.org), which renders clinical models 1–3 developed here accessible for the wider community of clinicians and researchers.

## Supporting information

**S1 Fig. Violin plot showing the distribution of liver fat percentage for the diabetes and non-diabetes cohorts of IMI DIRECT.**
(TIFF)

**S2 Fig. Distribution of liver fat percentage among the different centers contributing to the IMI DIRECT cohorts.**
(TIFF)

**S3 Fig. Manhattan plot showing SNPs associated with liver fat level (approximately 18 million imputed SNPs) in the IMI DIRECT cohorts.** The chromosomal position is plotted on the $x$-axis, and the statistical significance of association for each SNP is plotted on the $y$-axis. Red line indicates genome-wide significance level ($5 \times 10^{-8}$).
(TIFF)

**S4 Fig. Quantile–quantile (QQ) plot showing results of genome-wide association study (GWAS) for liver fat content in the IMI DIRECT consortium (1,514 individuals).** The $x$-axis illustrates the expected distribution of $p$-values from the association test across all SNPs, and the $y$-axis shows the observed $p$-values.
(TIFF)

**S5 Fig. Details of the feature selection step for models 4–14 and models 15–18 using the IMI DIRECT data.** Models 4–14 (blue box); models 15–18 (green box).
(TIFF)

**S6 Fig. Measurements of sensitivity, specificity, F1 score (a score considering sensitivity and precision combined), and balanced accuracy at different cutoffs for model 1 in the diabetes, non-diabetes, and combined cohorts of IMI DIRECT.**
(TIFF)

**S7 Fig. Measurements of sensitivity, specificity, F1 score (a score considering sensitivity and precision combined), and balanced accuracy at different cutoffs for model 2 in the diabetes, non-diabetes, and combined cohorts of IMI DIRECT.**
(TIFF)

**S8 Fig. Variable importance for the clinical model via a permutation accuracy importance measure.** The $y$-axis shows the top 20 predictors in the model. The $x$-axis shows the variable importance, calculated using random forest analysis as the difference in prediction accuracy before and after the permutation for each variable scaled by the standard error. ALT, alanine transaminase; AST, aspartate transaminase; BasalISR, insulin secretion at the beginning of the

OGTT/MMTT; BMI, body mass index; Clins, mean insulin clearance during the OGTT/MMTT calculated as (mean insulin secretion)/(mean insulin concentration); DBP, diastolic blood pressure; Diabetes_status2, non-diabetes/diabetes; GGTP, gamma-glutamyl transpeptidase; Glucagonmin0, fasting glucagon concentration; Glucose, fasting glucose from venous plasma samples; HbA1c, hemoglobin A1C; HDL, fasting high-density lipoprotein cholesterol; Insulin, fasting insulin from venous plasma samples; OGIS, oral glucose insulin sensitivity index according to the method of Mari et al. [24]; PA_intensity_mean, mean high-pass-filtered vector magnitude physical activity intensity; SBP, systolic blood pressure; TG, fasting triglycerides; TotGLP1min0, concentration of fasting total GLP-1 in plasma; TwoGlucose, 2-hour glucose after OGTT/MMTT; TwoInsulin, 2-hour insulin.
(TIFF)

**S9 Fig. Variable importance for the genetic model via a permutation accuracy importance measure.** The *y*-axis shows the top 20 predictors in the model. The *x*-axis shows the variable importance, calculated using random forest analysis as the difference in prediction accuracy before and after the permutation for each variable scaled by the standard error.
(TIFF)

**S10 Fig. Variable importance for the transcriptomic model via a permutation accuracy importance measure.** The *y*-axis shows the top 20 predictors in the model. The *x*-axis shows the variable importance, calculated using random forest analysis as the difference in prediction accuracy before and after the permutation for each variable scaled by the standard error.
(TIFF)

**S11 Fig. Variable importance for the exploratory proteomic model via a "permutation accuracy importance" measure.** The *y*-axis shows the top 20 predictors in the model. The *x*-axis shows the variable importance, calculated using random forest as the difference in prediction accuracy before and after the permutation for each variable scaled by the standard error.
(TIFF)

**S12 Fig. Variable importance for the targeted metabolomic model via a permutation accuracy importance measure.** The *y*-axis shows the top 20 predictors in the model. The *x*-axis shows the variable importance, calculated using random forest analysis as the difference in prediction accuracy before and after the permutation for each variable scaled by the standard error.
(TIFF)

**S13 Fig. Variable importance for the targeted proteomic model via a permutation accuracy importance measure.** The *y*-axis shows the top 20 predictors in the model. The *x*-axis shows the variable importance, calculated using random forest analysis as the difference in prediction accuracy before and after the permutation for each variable scaled by the standard error.
(TIFF)

**S14 Fig. Variable importance for the untargeted metabolomic model via a permutation accuracy importance measure.** The *y*-axis shows the top 20 predictors in the model. The *x*-axis shows the variable importance, calculated using random forest analysis as the difference in prediction accuracy before and after the permutation for each variable scaled by the standard error.
(TIFF)

**S15 Fig. Variable importance for the clinical model derived from ensemble feature selection (EFS).** The *y*-axis shows the 22 clinical variables ordered by importance value. The *x*-axis shows

the cumulative importance values, calculated via an ensemble of feature selection methods including Spearman's rank correlation test (S_cor), Pearson's product moment correlation test (P_cor), beta-values of logistic regression (LogReg), error-rate-based variable importance measure (ER_RF), and Gini-index-based variable importance measure (Gini_RF). ALT, alanine transaminase; AST, aspartate transaminase; BasalISR, insulin secretion at the beginning of the OGTT/MMTT; BMI, body mass index; Clins, mean insulin clearance during the OGTT/MMTT calculated as (mean insulin secretion)/(mean insulin concentration); DBP, diastolic blood pressure; Diabetes status, non-diabetes/diabetes; GGTP, gamma-glutamyl transpeptidase; Glucagonmin0, fasting glucagon concentration; Glucose, fasting glucose from venous plasma samples; HbA1c, hemoglobin A1C; HDL, fasting high-density lipoprotein cholesterol; Insulin, fasting insulin from venous plasma samples; OGIS, oral glucose insulin sensitivity index according to the method of Mari et al. [24]; PA_intensity_mean, mean high-pass-filtered vector magnitude physical activity intensity; SBP, systolic blood pressure; TG, fasting triglycerides; TotGLP1min0, concentration of fasting total GLP-1 in plasma; TwoGlucose, 2-hour glucose after OGTT/MMTT; TwoInsulin, 2-hour insulin.
(TIF)

**S16 Fig. Variable importance for the genetic model derived from EFS.** The *y*-axis shows the 23 genetic variables ordered by importance value. The *x*-axis shows the cumulative importance values, calculated via an ensemble of feature selection methods including Spearman's rank correlation test (S_cor), Pearson's product moment correlation test (P_cor), beta-values of logistic regression (LogReg), error-rate-based variable importance measure (ER_RF), and Gini-index-based variable importance measure (Gini_RF).
(TIFF)

**S17 Fig. Variable importance for the transcriptomic model derived from EFS.** The *y*-axis shows the 93 transcriptomic variables ordered by importance value. The *x*-axis shows the cumulative importance values, calculated via an ensemble of feature selection methods including Spearman's rank correlation test (S_cor), Pearson's product moment correlation test (P_cor), beta-values of logistic regression (LogReg), error-rate-based variable importance measure (ER_RF), and Gini-index-based variable importance measure (Gini_RF).
(TIFF)

**S18 Fig. Variable importance for the exploratory proteomic model derived from EFS.** The *y*-axis shows the 22 exploratory proteomic variables ordered by importance value. The *x*-axis shows the cumulative importance values, calculated via an ensemble of feature selection methods including Spearman's rank correlation test (S_cor), Pearson's product moment correlation test (P_cor), beta-values of logistic regression (LogReg), error-rate-based variable importance measure (ER_RF), and Gini-index-based variable importance measure (Gini_RF).
(TIFF)

**S19 Fig. Variable importance for the targeted metabolomic model derived from EFS.** The *y*-axis shows the 25 targeted metabolomic variables ordered by importance value. The *x*-axis shows the cumulative importance values, calculated via an ensemble of feature selection methods including Spearman's rank correlation test (S_cor), Pearson's product moment correlation test (P_cor), beta-values of logistic regression (LogReg), error-rate-based variable importance measure (ER_RF), and Gini-index-based variable importance measure (Gini_RF).
(TIFF)

**S20 Fig. Variable importance for the clinical plus multi-omics model (clinical = 22, genetic = 23, transcriptomic = 93, exploratory proteomic = 22, and targeted metabolomic = 25) derived from EFS.** The *y*-axis shows the top 20 predictors in the model. The *x*-axis shows the cumulative importance values, calculated via an ensemble of feature selection methods including Spearman's rank correlation test (S_cor), Pearson's product moment correlation test (P_cor), beta-values of logistic regression (LogReg), error-rate-based variable importance measure (ER_RF), and Gini-index-based variable importance measure (Gini_RF). ALT, alanine transaminase; AST, aspartate transaminase; BasalISR, insulin secretion at the beginning of the OGTT/MMTT; Clins, mean insulin clearance during the OGTT/MMTT calculated as (mean insulin secretion)/(mean insulin concentration); Insulin, fasting insulin from venous plasma samples; OGIS, oral glucose insulin sensitivity index according to the method of Mari et al. [24]; TG, fasting triglycerides; TotGLP1min0, concentration of fasting total GLP-1 in plasma; TwoInsulin, 2-hour insulin after OGTT/MMTT.
(TIFF)

**S1 STROBE Checklist. The STROBE (Strengthening the Reporting of Observational Studies in Epidemiology) checklist.**
(DOCX)

**S1 Table. The list of the clinical input variables with the abbreviation used in the analyses and their meaning.**
(XLSX)

**S2 Table. Characteristics of the study in the non-diabetes, diabetes, and combined cohorts separated for participants from IMI DIRECT who had MRI data versus those who did not have MRI data.** Values are median (interquartile range) unless otherwise specified. ALT, alanine transaminase; AST, aspartate transaminase; BMI, body mass index; DBP, diastolic blood pressure; HbA1c, hemoglobin A1C; SBP, systolic blood pressure.
(XLSX)

**S3 Table. Variables used to construct each of the NAFLD prediction models developed in IMI DIRECT.** ALT, alanine transaminase; AST, aspartate transaminase; BMI, body mass index; DBP, diastolic blood pressure; GGTP, gamma-glutamyl transpeptidase; Glucagonmin0, fasting glucagon concentration; HbA1c, hemoglobin A1C; HDL, fasting high-density lipoprotein cholesterol; MMTT, mixed meal tolerance test; OGIS, oral glucose insulin sensitivity index according to the method of Mari et al. [24]; OGTT, oral glucose tolerance test; PA_intensity_mean, mean high-pass-filtered vector magnitude physical activity intensity; SBP, systolic blood pressure; TotGLP1min0, concentration of fasting total GLP-1 in plasma.
(XLSX)

**S4 Table. UK Biobank field number with the description used in the analyses.**
(XLSX)

**S5 Table. Receiver operating characteristic area under the curve (ROCAUC) with 95% confidence interval for clinical models 1–3, fatty liver index (FLI), hepatic steatosis index (HSI), and NAFLD liver fat score (NAFLD-LFS) in the non-diabetes and diabetes cohorts of the IMI DIRECT separately.**
(XLSX)

**S6 Table. Receiver operating characteristic area under the curve (ROCAUC) with 95% confidence interval of each separate dataset obtained from random forest (RF), generalized linear model (GLM), stochastic gradient boosting (GBM), support vector machine (SVM),**

and *k*-nearest neighbor (KNN) analyses in the cross-validated test data of the IMI DIRECT combined cohort.
(XLSX)

**S7 Table. The details of the LASSO (least absolute shrinkage and selection operator)–selected features of the omics layers in separate sheets (genetic, transcriptomic, exploratory proteomic, targeted proteomic, targeted Metabolomic, and untargeted metabolomic).**
(XLSX)

**S1 Text. QC of the transcriptomic, proteomic, and metabolomic variables in the IMI DIRECT datasets.**
(DOCX)

**S1 TRIPOD Checklist. The TRIPOD (Transparent Reporting of a Multivariable Prediction Model for Individual Prognosis or Diagnosis) checklist.**
(DOCX)

## Acknowledgments

We thank Mattias Borell for developing, logistical support, and advice related to the web interface. We thank all the participants and study center staff in IMI DIRECT for their contribution to the study. We thank all the participants in the UK Biobank. This research was conducted using the UK Biobank resource (application ID: 18274). For the proteomic analyses, we thank the entire staff of the Human Protein Atlas, the Plasma Profiling Facility at Science for Life Laboratory, and in particular Elin Birgersson, Annika Bendes, and Eni Andersson for technical assistance. We thank C. Prehn (HMGU) for laboratory work related to the metabolomic data.

## Author Contributions

**Conceptualization:** Naeimeh Atabaki-Pasdar, Mattias Ohlsson, Jimmy D. Bell, Imre Pavo, Paul W. Franks.

**Data curation:** Naeimeh Atabaki-Pasdar, Ana Viñuela, Mark Haid, Angus G. Jones, E. Louise Thomas, Robert W. Koivula, Azra Kurbasic, Juan Fernandez, Adem Y. Dawed, Ian M. Forgie, Timothy J. McDonald, Matilda Dale, Federico De Masi, Mun-Gwan Hong, Tarja Kokkola, Helle Krogh Pedersen, Anubha Mahajan, Sapna Sharma, Jerzy Adamski, Soren Brage, Søren Brunak, Emmanouil Dermitzakis, Gary Frost, Andrea Mari, Jochen M. Schwenk, Ramneek Gupta, Jimmy D. Bell.

**Formal analysis:** Naeimeh Atabaki-Pasdar, Ana Viñuela, Francesca Frau, Hugo Pomares-Millan.

**Funding acquisition:** Jerzy Adamski, Søren Brunak, Torben Hansen, Markku Laakso, Oluf Pedersen, Martin Ridderstråle, Hartmut Ruetten, Andrew T. Hattersley, Mark Walker, Mark I. McCarthy, Ewan R. Pearson, Imre Pavo, Paul W. Franks.

**Investigation:** Robert W. Koivula, Timothy J. McDonald, Femke Rutters, Henna Cederberg, Kristine H. Allin, Alison Heggie, Gwen Kennedy, Henrik Vestergaard, Soren Brage, Søren Brunak, Torben Hansen, Markku Laakso, Oluf Pedersen, Martin Ridderstråle, Hartmut Ruetten, Andrew T. Hattersley, Mark Walker, Joline W. J. Beulens, Ramneek Gupta, Mark I. McCarthy, Ewan R. Pearson, Imre Pavo, Paul W. Franks.

**Methodology:** Naeimeh Atabaki-Pasdar, Mattias Ohlsson, Ana Viñuela, Pascal M. Mutie, Hugo Fitipaldi, Giuseppe N. Giordano, Elizaveta Chabanova, Cecilia Engel Thomas, Tue H. Hansen, Petra J. M. Elders, Donna McEvoy, Francois Pattou, Violeta Raverdy, Ragna S. Häussler, Henrik S. Thomsen, Leen M. 't Hart, Petra B. Musholt, Andrea Mari, Jochen M. Schwenk, Jimmy D. Bell, Paul W. Franks.

**Project administration:** Robert W. Koivula, Giuseppe N. Giordano, Ian M. Forgie, Tue H. Hansen, Gwen Kennedy, Jerzy Adamski, Søren Brunak, Torben Hansen, Markku Laakso, Oluf Pedersen, Martin Ridderstråle, Hartmut Ruetten, Andrew T. Hattersley, Mark Walker, Joline W. J. Beulens, Jochen M. Schwenk, Mark I. McCarthy, Ewan R. Pearson, Jimmy D. Bell, Paul W. Franks.

**Resources:** Federico De Masi, Søren Brunak, Paul W. Franks.

**Software:** Naeimeh Atabaki-Pasdar, Paul W. Franks.

**Supervision:** Mattias Ohlsson, Paul W. Franks.

**Visualization:** Naeimeh Atabaki-Pasdar.

**Writing – original draft:** Naeimeh Atabaki-Pasdar, Paul W. Franks.

**Writing – review & editing:** Naeimeh Atabaki-Pasdar, Mattias Ohlsson, Ana Viñuela, Francesca Frau, Hugo Pomares-Millan, Mark Haid, Angus G. Jones, E. Louise Thomas, Robert W. Koivula, Azra Kurbasic, Pascal M. Mutie, Hugo Fitipaldi, Juan Fernandez, Adem Y. Dawed, Giuseppe N. Giordano, Ian M. Forgie, Timothy J. McDonald, Femke Rutters, Henna Cederberg, Elizaveta Chabanova, Matilda Dale, Federico De Masi, Cecilia Engel Thomas, Kristine H. Allin, Tue H. Hansen, Alison Heggie, Mun-Gwan Hong, Petra J. M. Elders, Gwen Kennedy, Tarja Kokkola, Helle Krogh Pedersen, Anubha Mahajan, Donna McEvoy, Francois Pattou, Violeta Raverdy, Ragna S. Häussler, Sapna Sharma, Henrik S. Thomsen, Jagadish Vangipurapu, Henrik Vestergaard, Leen M. 't Hart, Jerzy Adamski, Petra B. Musholt, Soren Brage, Søren Brunak, Emmanouil Dermitzakis, Gary Frost, Torben Hansen, Markku Laakso, Oluf Pedersen, Martin Ridderstråle, Hartmut Ruetten, Andrew T. Hattersley, Mark Walker, Joline W. J. Beulens, Andrea Mari, Jochen M. Schwenk, Ramneek Gupta, Mark I. McCarthy, Ewan R. Pearson, Jimmy D. Bell, Imre Pavo, Paul W. Franks.

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
