## [Editor Report · Decision Letter 0]

21 Jan 2020

Dear Dr Franks, 

Thank you for submitting your manuscript entitled "Predicting and elucidating the etiology of fatty liver disease using a machine learning-based approach: an IMI DIRECT study" for consideration by PLOS Medicine.

Your manuscript has now been evaluated by the PLOS Medicine editorial staff and I am writing to let you know that we would like to send your submission out for external peer review.

Kind regards,

Helen Howard, for Clare Stone PhD 

Acting Editor-in-Chief

PLOS Medicine 

plosmedicine.org

---

## [Decision Letter · Decision Letter 1]

19 Feb 2020

Dear Dr. Franks,

Thank you very much for submitting your manuscript "Predicting and elucidating the etiology of fatty liver disease using a machine learning-based approach: an IMI DIRECT study" (PMEDICINE-D-20-00136R1) for consideration at PLOS Medicine. 

[LINK]

In light of these reviews, I am afraid that we will not be able to accept the manuscript for publication in the journal in its current form, but we would like to consider a revised version that addresses the reviewers' and editors' comments. Obviously we cannot make any decision about publication until we have seen the revised manuscript and your response, and we plan to seek re-review by one or more of the reviewers. 

We expect to receive your revised manuscript by Mar 11 2020 11:59PM. Please email us (plosmedicine@plos.org) if you have any questions or concerns.

We look forward to receiving your revised manuscript. 

Sincerely,

Caitlin Moyer, Ph.D.

Associate Editor 

PLOS Medicine

plosmedicine.org

Ref 3 raises some significant points and we would like to see all of these issues resolved in revisions. 

Other editorial points: 

Title - Please revise your title according to PLOS Medicine's style. Your title must be nondeclarative and not a question. It should begin with main concept if possible. "Effect of" should be used only if causality can be inferred, i.e., for an RCT. Please place the study design ("A randomized controlled trial," "A retrospective study," "A modelling study," etc.) in the subtitle (ie, after a colon). Ther study design should be a machine learning approach.

Abstract – (here and elsewhere including main text and tables) please ensure that p values are provided where 95%Cis are given. Please also include a sentence or two on the study’s limitations as the final part of the Methods and Findings section. Please also add summary demographic information to the abstract and please also mention the 2 cohorts from and where they are recruited from. 

Please use square brackets for refs in the main text instead of curved. 

Please ensure that the study is reported according to the [STROBE] guideline (for the cohort aspect), and include the completed [STROBE or other] checklist as Supporting Information. When completing the checklist, please use section and paragraph numbers, rather than page numbers. Please add the following statement, or similar, to the Methods: "This study is reported as per the Strengthening the Reporting of Observational Studies in Epidemiology (STROBE) guideline (S1 Checklist)."

Please report your study according to the relevant guideline, which can be found here: http://www.equator-network.org/

 transparent reporting of a multivariable prediction model for individual prognosis or diagnosis (TRIPOD) guideline (S1 Checklist).

Comments from the reviewers:

Reviewer #1: This study describes a NAFLD prediction model based on multi-omic and clinical data using machine learning. The topic is of great interenst and the manuscript is overall well presented.

Reviewer #2: This is a very interesting study with significant implications.

1] The link between NAFLD and cardiovascular disease (CVD) should be briefly mentioned since it further highlights the importance of the early detection and treatment of NAFLD.

Useful refs:

Targher G, Byrne CD, Lonardo A, Zoppini G, Barbui C. Non-alcoholic fatty liver disease and risk of incident cardiovascular disease: A meta-analysis. J Hepatol. 2016;65(3):589-600.

Mahfood Haddad T, Hamdeh S, Kanmanthareddy A, Alla VM. Nonalcoholic fatty liver disease and the risk of clinical cardiovascular events: A systematic review and meta-analysis. Diabetes Metab Syndr. 2017;11 Suppl 1:S209-S216.

2] Please comment on other relevant papers such as:

Perakakis N, Polyzos SA, Yazdani A, Sala-Vila A, Kountouras J, Anastasilakis AD, Mantzoros CS. Non-invasive diagnosis of non-alcoholic steatohepatitis and fibrosis with the use of omics and supervised learning: A proof of concept study. Metabolism. 2019;101:154005. 

Katsiki N, Gastaldelli A, Mikhailidis DP. Predictive models with the use of omics and supervised machine learning to diagnose non-alcoholic fatty liver disease: A "non-invasive alternative" to liver biopsy? Metabolism. 2019;101:154010. 

3] What about the cost of equipments, stuff, measurements etc for this machine-learning approach? What about the complexity of the procedure?

In other words, is it possible to perform this technique in daily clinical practice? 

How can we suggest that this approach is cost-effective? Are there any data?

Reviewer #3: This study built several machine-learning models for predicting non-alcoholic fatty liver disease based on clinical, genetic, transcriptomic, proteomic, and metabolomic data. The authors also built a nice website tool for clinicians to easily use their prediction models. 

The authors tried to integrate multiple sources of data, however, the authors overlooked the innate differences of omics and clinical data in data type (discrete, continuous, categorical), scale and distribution. There is no data harmonization or normalization between different data source and simply putting them into one single prediction model and interpreting the importance ranking can be misleading. In addition, many studies have revealed correlation between omics data, for example, genetic and transcriptomic data (i.e., eQTLs). However, the authors only applied LASSO within each omics data and did not discuss the potential collinearity issues between the selected omics features. One possible solution is that, instead of putting all features from multiple data sources in to one model, the authors can use boosting algorithms that ensemble the prediction model for each data source into one stronger prediction model. 

Other several minor aspects that the authors can improve:

1. The flowchart is quite complicated. One suggestion is to build separate flowcharts for feature selection and model building, or at least put them in order so that the readers can clearly follow the steps. 

2. The QQ plot for the GWAS in the supplementary figure looks abnormal at the top. Please check what variables are used to conduct the GWAS and include all possible confounders. 

3. The authors can expand details on what type of data standardization was done within each data source. Please clarify how "inverse normalized" was applied. 

4. While random forest is a useful machine learning model for classification, the authors can try at least one other method (e.g. XGBoost, support vector machines, k-nearest neighbor) and select the best performing models.

Reviewer #4: The paper is well written and the objectives are clear. Prediciton of NAFLD is an important aspect in for diagnostics, e.g., in case of the metabolic syndrome. I have only some minor comments:

1. Recently, Canbay et al. published a machine learning approach for the prediction of NAFLD with a similar AUC, with partly overlapping parameters, e.g., liver parameters. This paper should be discussed and compared to the findings in this study and the advantage of the new study should be clearly described.

2. The authors used mean decrease in accuracy as a measure of variable importance. However, this measure has been shown to be biased, see e.g., pubmed:17254353 or pubmed:23560875. An unbiased alternative could be the use of ensemble feature selection (EFS), see e.g., pubmed:28674556. I recommend to run an EFS selection, to compare the selected features with those from the RFs, and to discuss this at least in the discussion.

[LINK]

---

## [Decision Letter · Decision Letter 2]

5 May 2020

Dear Dr. Franks,

Thank you very much for re-submitting your manuscript "Predicting and elucidating the etiology of fatty liver disease: a machine learning modelling and validation study in the IMI DIRECT cohorts" (PMEDICINE-D-20-00136R2) for review by PLOS Medicine.

I have discussed the paper with my colleagues and the academic editor and it was also seen again by two reviewers. I am pleased to say that provided the remaining editorial and production issues are dealt with we are planning to accept the paper for publication in the journal.

[LINK]

We look forward to receiving the revised manuscript by May 12 2020 11:59PM. 

Sincerely,

Caitlin Moyer, Ph.D.

Associate Editor 

PLOS Medicine

plosmedicine.org

Requests from Editors:

1.The Data Availability Statement (DAS) requires revision. It is too vague to state that “Data cannot be shared publicly because GDPR restrictions on data privacy.”

Please see the policy at 

http://journals.plos.org/plosmedicine/s/data-availability

and FAQs at 

http://journals.plos.org/plosmedicine/s/data-availability#loc-faqs-for-data-policy

We suggest the following, or similar: “Data cannot be shared publicly due to a need to maintain the confidentiality of patient data. Interested researchers may contact [please provide a web link or email address, note that this cannot be a study author] to request and obtain relevant data.”

2.Abstract: Lines 6-7: The final sentence of the Abstract’s Background should clearly state the study question. We suggest moving this sentence to the last sentence of the Abstract’s Background section: “We sought to expand etiological understanding and develop a diagnostic tool for NAFLD using machine learning.”

3.Abstract and throughout: We appreciate your previous response regarding the inclusion of p values, however please include p values to accompany all 95% CIs reported throughout the text. Please report very small p values as p<0.001 where applicable.

4.Abstract: Methods and Findings: In the last sentence of the Abstract Methods and Findings section, please describe the main limitation(s) of the study's methodology.

5.Abstract: Conclusions: Please interpret the study based on the results presented in the abstract, emphasizing what is new without overstating your conclusions, particularly as your study does not test clinical use of the new model. Please mention specific implications substantiated by the results. The phrase "In this study, we found..." may be useful. Please revise the text at line 24-25 to avoid any causal implications, we suggest: “...identified biological features that appear to be associated with liver fat accumulation.” or similar.

6.Author Summary: Thank you for providing an Author Summary. Please consider combining some of the bullet points, for example:

“Why was this study done”: We suggest you combine the first two bullet points, and the 4th and 5th bullet points, below is a suggestion: 

“Globally, about one in four adults have NAFLD, which adversely affects energy homeostasis (particular blood glucose concentrations), blood detoxification, drug metabolism and food digestion.”

“The purpose of this work was to develop accurate non-invasive methods to aid in the clinical prediction of NAFLD”

7.Introduction: Line 77: Please refer to high income countries rather than "Western" countries.

8.Methods: Line 95 and 109 and Table 1 (and throughout): Please replace “diabetics” with “persons with diabetes.” Similarly, please consider revising use of the terms “fatty” and “non-fatty” to describe individuals in the cohort.

9.Methods: Line 96: Please specify whether informed consent was written or oral.

10.Methods: Lines 100-121: Please include somewhere in this paragraph a sentence describing or add the fraction of individuals for whom the MRI data were available relative to the total number in the cohort.

11.Methods: Lines 148-152: Please remove the use of italics.

12.Methods: Line 155-159: Thanks for discussing some limitations of the study, please move this discussion to the Discussion section of the manuscript.

13.Results: Line 334, and throughout: Please provide p values in addition to 95% CIs, and please report p values as p<0.001 where applicable.

14.Results: Line 465-467: The description of the web interface would be more appropriate in the Discussion section.

15.Results: Line 469-470: Please move the sentence regarding the TRIPOD guidelines to the Methods section.

16.Discussion: Line 476-477: Please modify this sentence to identify which models, or how many of the 18, were validated: “The models were successfully validated in the UK Biobank, where data permitted”

17.Discussion: Line 540: Please clarify the sentence “The models developed here may be used for screening.” to “The models developed here may be useful for screening for NAFLD and this should be evaluated in future clinical studies”, or similar.

18. Discussion: Please present and organize the Discussion as follows: a short, clear summary of the article's findings; what the study adds to existing research and where and why the results may differ from previous research; strengths and limitations of the study; implications and next steps for research, clinical practice, and/or public policy; one-paragraph conclusion. In particular, please make the discussion of strengths and limitations more clearly defined.

19. Table 1: In the legend, please define abbreviations for BMI, SBP, DBP, HbA1c, ALC and AST.

20.Figure 1: Please make sure the abbreviations used in the figure are defined in the legend that will accompany this figure. Please also provide a label for the color map used.

21. Figures and Tables: Please make sure that descriptive titles and legends are included for each figure and table (including those in Supporting Information files). Please make sure that all abbreviations used in each figure and table are defined in that figure/table legend.

22. S2 Checklist: Thank you for including the TRIPOD checklist, please revise the checklist using section and paragraphs to refer to locations within the manuscript, rather than page numbers.

Comments from Reviewers:

Reviewer #2: The comments have been adequately responded

Reviewer #4: The authors addressed all my questions sufficiently.

[LINK]

---

## [Editor Report · Decision Letter 3]

22 May 2020

Dear Prof. Franks, 

On behalf of my colleagues and the academic editor, Dr. Dominik Heider, I am delighted to inform you that your manuscript entitled "Predicting and elucidating the etiology of fatty liver disease: a machine learning modelling and validation study in the IMI DIRECT cohorts" (PMEDICINE-D-20-00136R3) has been accepted for publication in PLOS Medicine. 

PRODUCTION PROCESS

PRESS

PROFILE INFORMATION

Thank you again for submitting the manuscript to PLOS Medicine. We look forward to publishing it. 

Best wishes, 

Caitlin Moyer, Ph.D.

Associate Editor 

PLOS Medicine

plosmedicine.org